# Understanding How Encoder-Decoder Architectures Attend

**Kyle Aitken**
Department of Physics
University of Washington
Seattle, Washington, USA
kaitken17@gmail.com

**Vinay V Ramasesh**
Google Research, Blueshift Team
Mountain View, California, USA

**Yuan Cao**
Google, Inc.
Mountain View, California, USA

**Niru Maheswaranathan**
Google Research, Brain Team
Mountain View, California, USA

## Abstract

Encoder-decoder networks with attention have proven to be a powerful way to solve many sequence-to-sequence tasks. In these networks, attention aligns encoder and decoder states and is often used for visualizing network behavior. However, the mechanisms used by networks to generate appropriate attention matrices are still mysterious. Moreover, how these mechanisms vary depending on the particular architecture used for the encoder and decoder (recurrent, feed-forward, etc.) are also not well understood. In this work, we investigate how encoder-decoder networks solve different sequence-to-sequence tasks. We introduce a way of decomposing hidden states over a sequence into *temporal* (independent of input) and *input-driven* (independent of sequence position) components. This reveals how attention matrices are formed: depending on the task requirements, networks rely more heavily on either the temporal or input-driven components. These findings hold across both recurrent and feed-forward architectures despite their differences in forming the temporal components. Overall, our results provide new insight into the inner workings of attention-based encoder-decoder networks.

## 1 Introduction

Modern machine learning encoder-decoder architectures can achieve strong performance on sequence-to-sequence tasks such as machine translation (Bahdanau et al., 2014; Luong et al., 2015; Wu et al., 2016; Vaswani et al., 2017), language modeling (Raffel et al., 2020), speech-to-text (Chan et al., 2015; Prabhavalkar et al., 2017; Chiu et al., 2018), etc. Many of these architectures make use of attention (Bahdanau et al., 2014), a mechanism that allows the network to focus on a specific part of the input most relevant to the current prediction step. Attention has proven to be a critical mechanism; indeed many modern architectures, such as the Transformer, are fully attention-based (Vaswani et al., 2017). However, despite the success of these architectures, an understanding of *how* said networks solve such tasks using attention remains largely unknown.

Attention mechanisms are attractive because they are interpretable, and often illuminate key computations required for a task. For example, consider neural machine translation—trained networks exhibit attention matrices that align words in the encoder sequence with the appropriate corresponding position in the decoder sentence (Ghader & Monz, 2017; Ding et al., 2019). In this case, the attention matrix already contains information about which words in the source sequence are relevant for translating a particular word in the target sequence; that is, forming the attention matrix itself

35th Conference on Neural Information Processing Systems (NeurIPS 2021).

constitutes a significant part of solving the overall task. How is it that networks are able to achieve this? What are the mechanisms underlying how networks form attention, and how do they vary across tasks and architectures?

In this work, we study these questions by analyzing three different encoder-decoder architectures on sequence-to-sequence tasks. We develop a method for decomposing the hidden states of the network into a sum of components that let us isolate *input* driven behavior from *temporal* (or sequence) driven behavior. We use this to first understand how networks solve tasks where all samples use the same attention matrix, a diagonal one. We then build on that to show how additional mechanisms can generate sample-dependent attention matrices that are still close to the average matrix.

**Our Contributions**

- We propose a decomposition of hidden state dynamics into separate pieces, one of which explains the temporal behavior of the network, another of which describes the input behavior. We show such a decomposition aids in understanding the behavior of networks with attention.
- In the tasks studied, we show the temporal (input) components play a larger role in determining the attention matrix as the average attention matrix becomes a better (worse) approximation for a random sample's attention matrix.
- We discuss the dynamics of architectures with attention and/or recurrence and show how the input/temporal component behavior differs across said architectures.
- We investigate the detailed temporal and input component dynamics in a synthetic setting to understand the mechanism behind common sequence-to-sequence structures and how they might differ in the presence of recurrence.

**Related Work**   As mentioned in the introduction, a common technique to gain some understanding is to visualize learned attention matrices, though the degree to which such visualization can explain model predictions is disputed Wiegreffe & Pinter (2019); Jain & Wallace (2019); Serrano & Smith (2019). Input saliency Bastings & Filippova (2020) and attribution-propagation Chefer et al. (2020) methods have also been studied as potential tools for model interpretability.

Complementary to these works, our approach builds on a recent line of work analyzing the computational mechanisms learned by RNNs from a dynamical systems perspective. These analyses have identified simple and interpretable hidden state dynamics underlying RNN operation on text-classification tasks such as binary sentiment analysis (Maheswaranathan et al., 2019; Maheswaranathan & Sussillo, 2020) and document classification (Aitken et al., 2020). Our work extends these ideas into the domain of sequence-to-sequence tasks.

**Notation**   Let $T$ and $S$ be the input and output sequence length of a given sample, respectively. We denote the encoder and decoder hidden states by $\mathbf{h}_t^{\mathrm{E}} \in \mathbb{R}^n$ with $t = 1, \ldots, T$. Similarly, we denote decoder hidden states by $\mathbf{h}_s^{\mathrm{D}} \in \mathbb{R}^n$, with $s = 1, \ldots, S$. The encoder and decoder hidden state dimensions are always taken to be equal in this work. Inputs to the encoder and decoder are denoted by $\mathbf{x}_t^{\mathrm{E}} \in \mathbb{R}^d$ and $\mathbf{x}_s^{\mathrm{D}} \in \mathbb{R}^{\tilde{d}}$. When necessary, we subscript different samples from a test/train set using $\alpha, \beta, \gamma$, e.g. $\mathbf{x}_{t,\alpha}^{\mathrm{E}}$ for $\alpha = 1, \ldots, M$.

**Outline**   We begin by introducing the three architectures we investigate in this work with varying combinations of recursion and attention. Next we introduce our temporal and input component decomposition and follow this up with a demonstration of how such a decomposition allows us to understand the dynamics of attention in a simple one-to-one translation task. Afterwards, we apply this decomposition to two additional tasks with increasing levels of complexity and discuss how our decomposition gives insight into the behavior of attention in these tasks.

## 2   Setup

A schematic of the three architectures we study is shown in Fig. 1 (see SM for precise expressions).

**Vanilla Encoder Decoder (VED)** is a recurrent encoder-decoder architecture with no attention (Sutskever et al., 2014). The encoder and decoder update expression are $\mathbf{h}_t^{\mathrm{E}} = F_{\mathrm{E}}(\mathbf{h}_{t-1}^{\mathrm{E}}, \mathbf{x}_t^{\mathrm{E}})$ and $\mathbf{h}_s^{\mathrm{D}} = F_{\mathrm{D}}(\mathbf{h}_{s-1}^{\mathrm{D}}, \mathbf{x}_s^{\mathrm{D}})$, respectively. Here, $F_{\mathrm{D}}$ and $F_{\mathrm{E}}$ are functions that implement the hidden state updates, which in this work are each one of three modern RNN cells: LSTMs (Hochreiter & Schmidhuber, 1997), GRUs (Cho et al., 2014), or UGRNNs (Collins et al., 2016).

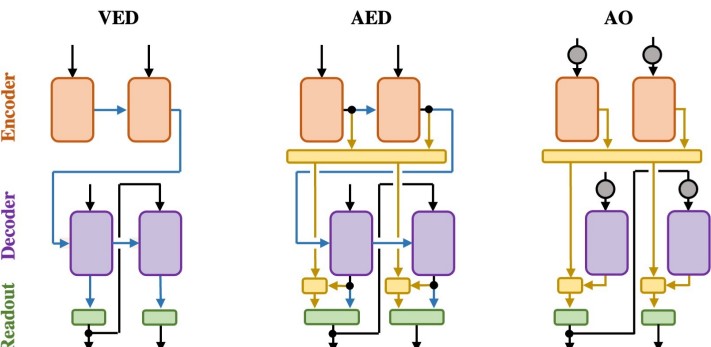

Figure 1: **Schematic of the three primary architectures analyzed in this work.** The orange, purple, and green boxes represent the encoder RNNs, decoder RNNs, and linear readout layers, respectively. Recurrent connections are shown in blue, attention-based connections and computational blocks are shown in gold. The grey circles add positional encoding to the inputs.

**Encoder-Decoder with Attention (AED)** is identical to the VED architecture above with a simple attention mechanism added (Bahdanau et al., 2014; Luong et al., 2015). For time step $s$ of the decoder, we compute a context vector $\mathbf{c}_s$, a weighted sum of encoder hidden states, $\mathbf{c}_s := \sum_{t=1}^{T} \alpha_{st} \mathbf{h}_t^{\mathrm{E}}$, with $\boldsymbol{\alpha}_t := \mathrm{softmax}\,(a_{1t}, \ldots, a_{St})$ the $t^{\mathrm{th}}$ column of the *attention matrix* and $a_{st} := \mathbf{h}_s^{\mathrm{D}} \cdot \mathbf{h}_t^{\mathrm{E}}$ the *alignment* between a given decoder and encoder hidden state. While more complicated attention mechanisms exist, in the main text we analyze the simplest form of attention for convenience of analysis.[1]

**Attention Only (AO)** is identical to the AED network above, but simply eliminates the recurrent information passed from one RNN cell to the next and instead adds fixed positional encoding vectors to the encoder and decoder inputs (Vaswani et al., 2017). Due to the lack of recurrence, the RNN functions $F_{\mathrm{E}}$ and $F_{\mathrm{D}}$ simply act as feedforward networks in this setting.[2] AO can be treated as a simplified version of a Transformer without self-attention, hence our analysis may also provide a hint into their inner workings (Vaswani et al., 2017).

## 2.1 Temporal and Input Components

In architectures with attention, we will show that it is helpful to write the hidden states using what we will refer to as their *temporal* and *input* components. This will be useful because each hidden state has an associated time step and input word at that same time step (e.g. $s$ and $\mathbf{x}_s^{\mathrm{D}}$ for $\mathbf{h}_s^{\mathrm{D}}$), therefore such a decomposition will often allow us to disentangle temporal and input behavior from any other network dynamics.

We define the temporal components of the encoder and decoder to be the average hidden state at a given time step, which we denote by $\boldsymbol{\mu}_t^{\mathrm{E}}$ and $\boldsymbol{\mu}_s^{\mathrm{D}}$, respectively. Similarly, we define an encoder input component to be the average of all $\mathbf{h}_t^{\mathrm{E}} - \boldsymbol{\mu}_t^{\mathrm{E}}$ for hidden states that immediately follow a given input word. We analogously define the decoder input components. In practice, we estimate such averages using a test set of size $M$, so that the temporal and input components of the encoder are respectively given by

$$\boldsymbol{\mu}_t^{\mathrm{E}} \approx \frac{\sum_{\alpha=1}^{M} \mathbf{1}_{\leq \mathrm{EoS},\alpha} \mathbf{h}_{t,\alpha}^{\mathrm{E}}}{\sum_{\beta=1}^{M} \mathbf{1}_{\leq \mathrm{EoS},\beta}}\,, \qquad \boldsymbol{\chi}^{\mathrm{E}}\left(\mathbf{x}_{t,\alpha}\right) \approx \frac{\sum_{\beta=1}^{M} \sum_{t'=1}^{T} \mathbf{1}_{\mathbf{x}_{t,\alpha},\mathbf{x}_{t',\beta}} \left(\mathbf{h}_{t',\beta}^{\mathrm{E}} - \boldsymbol{\mu}_{t'}^{\mathrm{E}}\right)}{\sum_{\gamma=1}^{M} \sum_{t''=1}^{T} \mathbf{1}_{\mathbf{x}_{t,\alpha},\mathbf{x}_{t'',\gamma}}}\,, \qquad (1)$$

where $\mathbf{h}_{t,\alpha}^{\mathrm{E}}$ the encoder hidden state of the $\alpha$th sample, $\mathbf{1}_{\leq \mathrm{EoS},\alpha}$ is a mask that is zero if the $\alpha$th sample is beyond the end of sentence, $\mathbf{1}_{\mathbf{x}_{t,\alpha},\mathbf{x}_{t',\beta}}$ is a mask that is zero if $\mathbf{x}_{t,\alpha} \neq \mathbf{x}_{t',\beta}$, and we have temporarily suppressed superscripts on the inputs for brevity.[3] By definition, *the temporal components only vary with time and the input components only vary with input/output word*. As such,

---

[1]In the SM, we implement a learned-attention mechanism using a scaled-dot product attention in the form of queries, keys, and value matrices (Vaswani et al., 2017). For the AED and AO architectures, we find qualitatively similar results to the simple dot-product attention presented in the main text.

[2]We train non-gated feedforward networks and find their dynamics to be qualitatively the same, see SM.

[3]See SM for more details on this definition and the analogous decoder definitions.

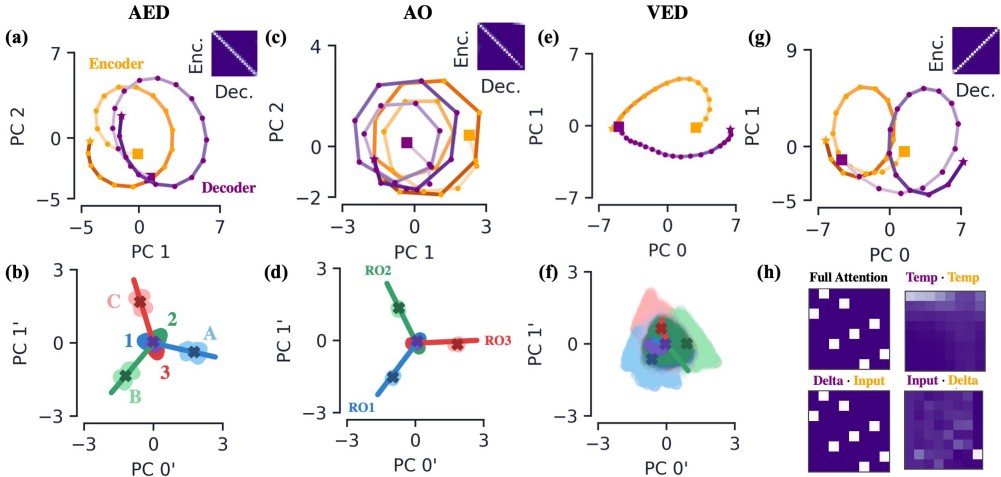

Figure 2: **Summary of attention dynamics on synthetic tasks.** **(a-f)** All three architectures trained on an $N = 3$ one-to-one translation task of variable length ranging from 15 to 20. Plots in the top row are projected onto the principal components (PCs) of the encoder and decoder temporal components, while those in the bottom row are projected onto the PCs of the input components. **(a)** For AED, the path formed by the temporal components of the encoder (orange) and decoder (purple), $\boldsymbol{\mu}_t^E$ and $\boldsymbol{\mu}_s^D$. We denote the first and last temporal component by a square and star, respectively, and the color of said path is lighter for earlier times. The inset shows the softmaxed alignment scores for $\boldsymbol{\mu}_s^D \cdot \boldsymbol{\mu}_t^E$, which we find to be a good approximation to the full alignment for the one-to-one translation task. **(b)** The input-delta components of the encoder (light) and decoder (dark) colored by word (see labels). The encoder input components, $\boldsymbol{\chi}_x^E$ are represented by a dark colored 'X'. The solid lines are the readout vectors (see labels on (d)). Start/end of sentence characters are in purple. **(c, d)** The same plots for the AO network. **(e, f)** The same plots for the VED network (with no attention inset). **(g)** Temporal components for the same task with a temporally reversed output sequence. **(h)** Attention matrices for a test example from a network trained to alphabetically sort a list of letters. Clockwise from top left, the softmaxed attention from the full hidden states ($\mathbf{h}_s^D \cdot \mathbf{h}_t^E$), temporal components only ($\boldsymbol{\mu}_s^D \cdot \boldsymbol{\mu}_t^E$), decoder input components and encoder delta components ($\boldsymbol{\chi}_y^D \cdot \Delta\mathbf{h}_t^E$), and decoder delta components and encoder input components ($\Delta\mathbf{h}_s^D \cdot \boldsymbol{\chi}_x^E$).

it will be useful to denote the encoder and decoder input components by $\boldsymbol{\chi}_x^E$ and $\boldsymbol{\chi}_y^D$, with $x$ and $y$ respectively running over all input and output words (e.g. $\boldsymbol{\chi}_{yes}^E$ and $\boldsymbol{\chi}_{oui}^D$). We can then write any hidden state as

$$\mathbf{h}_t^E = \boldsymbol{\mu}_t^E + \boldsymbol{\chi}_x^E + \Delta\mathbf{h}_t^E, \quad \mathbf{h}_s^D = \boldsymbol{\mu}_s^D + \boldsymbol{\chi}_y^D + \Delta\mathbf{h}_s^D, \tag{2}$$

with $\Delta\mathbf{h}_t^E := \mathbf{h}_t^E - \boldsymbol{\mu}_t^E - \boldsymbol{\chi}_x^E$ and $\Delta\mathbf{h}_s^D := \mathbf{h}_s^D - \boldsymbol{\mu}_s^D - \boldsymbol{\chi}_y^D$ the *delta components* of encoder and decoder hidden states, respectively. Intuitively, we are simply decomposing each hidden state vector as a sum of a component that only varies with time/position in the sequence (independent of input), a component that only varies with input (independent of position), and whatever else is left over. Finally, we will often refer to hidden states without their temporal component, i.e. $\boldsymbol{\chi}_x^E + \Delta\mathbf{h}_t^E$ and $\boldsymbol{\chi}_y^D + \Delta\mathbf{h}_s^D$, so for brevity we refer to these combinations as the *input-delta components*.

Using the temporal and input components in (2), we can decompose the attention alignment between two hidden states as

$$a_{st} = \left(\boldsymbol{\mu}_s^D + \boldsymbol{\chi}_y^D + \Delta\mathbf{h}_s^D\right) \cdot \left(\boldsymbol{\mu}_t^E + \boldsymbol{\chi}_x^E + \Delta\mathbf{h}_t^E\right). \tag{3}$$

We will show below that in certain cases several of the nine terms of this expression approximately vanish, leading to simple and interpretable attention mechanisms.

## 3 One-to-One Results

To first establish a basis of how each of the three architectures learn to solve tasks and the role of their input and temporal components, we start by studying their dynamics for a synthetic one-to-one translation task. The task is to convert a sequence of input words into a corresponding sequence of output words, where there is a one-to-one translation dictionary, e.g. converting a sequence of letters to their corresponding position in the alphabet, $\{B, A, C, A, D\} \rightarrow \{2, 1, 3, 1, 4\}$. We generate the

input phrases to have variable length, but outputs always have equal length to their input (i.e. $T = S$). While a solution to this task is trivial, it is not obvious how each neural network architecture will solve the task. Although this is a severely simplified approximation to realistic sequence-to-sequence tasks, we will show below that many of the dynamics the AED and AO networks learn on this task are qualitatively present in more complex tasks.

**Encoder-Decoder with Attention.** After training the AED architecture, we apply the decomposition of (2) to the hidden states. Plotting the temporal components of both the encoder and decoder, they each form an approximate circle that is traversed as their respective inputs are read in (Fig. 2a).[4] Additionally, we find the encoder and decoder temporal components are closest to alignment when $s = t$. We also plot the input components of the encoder and decoder together with the encoder input-delta components, i.e. $\boldsymbol{\chi}_x^{\mathrm{E}} + \Delta\mathbf{h}_t^{\mathrm{E}}$, and the network's readout vectors (Fig. 2b).[5] We see for the encoder hidden states, the input-delta components are clustered close to their respective input components, meaning for this task the delta components are negligible. Also note the decoder input-delta components are significantly smaller in magnitude than the decoder temporal components. Together, this means we can approximate the encoder and decoder hidden states as $\mathbf{h}_t^{\mathrm{E}} \approx \boldsymbol{\mu}_t^{\mathrm{E}} + \boldsymbol{\chi}_x^{\mathrm{E}}$ and $\mathbf{h}_s^{\mathrm{D}} \approx \boldsymbol{\mu}_s^{\mathrm{D}}$, respectively. Finally, note the readout vector for a given output word aligns with the input components of its translated input word, e.g. the readout for '1' aligns with the input component for 'A' (Fig. 2b).[6]

For the one-to-one translation task, the network learns an approximately diagonal attention matrix, meaning the decoder at time $s$ primarily attends to the encoder's hidden state at $t = s$. Additionally, we find the temporal and input-delta components to be close to orthogonal for all time steps, which allows the network's attention mechanism to isolate temporal dependence rather than input dependence. Since we can approximate the hidden states as $\mathbf{h}_t^{\mathrm{E}} \approx \boldsymbol{\mu}_t^{\mathrm{E}} + \boldsymbol{\chi}_x^{\mathrm{E}}$ and $\mathbf{h}_s^{\mathrm{D}} \approx \boldsymbol{\mu}_s^{\mathrm{D}}$, and the temporal and input components are orthogonal, the alignment in (3) can be written simply as $a_{st} \approx \boldsymbol{\mu}_s^{\mathrm{D}} \cdot \boldsymbol{\mu}_t^{\mathrm{E}}$. This means that the *full* attention is completely described by the temporal components and thus input-independent (this will not necessarily be true for other tasks, as we will see later).

With the above results, we can understand how AED solves the one-to-one translation task. After reading a given input, the encoder hidden state is primarily composed of an input and temporal component that are approximately orthogonal to one another, with the input component aligned with the readout of the translated input word (Fig. 2b). The decoder hidden states are approximately made up of only a temporal component, whose sole job is to align with the corresponding encoder temporal component. Temporal components of the decoder and encoder are closest to alignment for $t = s$, so the network primarily attends to the encoder state $\mathbf{h}_{t=s}^{\mathrm{E}}$. The alignment between encoder input components and readouts yields maximum logit values for the correct translation.

**Attention Only.** Now we turn to AO architecture, which is identical to AED except with the recurrent connections cut, and positional encoding added to the inputs. We find that AO has qualitatively similar temporal components that give rise to diagonal attention (Fig. 2c) and the input components align with the readouts (Fig. 2d). Thus AO solves the task in a similar manner as AED. The only difference is that the temporal components, driven by RNN dynamics in AED, are now driven purely by the positional encoding in AO.

**Vanilla Encoder-Decoder.** After training the VED architecture, we find the encoder and decoder hidden states belonging to the same time step form clusters, and said clusters are closest to those corresponding to adjacent time steps. This yields temporal components that are close to one another for adjacent times, with $\boldsymbol{\mu}_T^{\mathrm{E}}$ next to $\boldsymbol{\mu}_1^{\mathrm{D}}$ (Figs. 2e). Since there is no attention in this architecture, there is no incentive for the network to align temporal components of the encoder and decoder as we saw in AED and AO.

---

[4]Here and in plots that follow, we plot the various components using principal component analysis (PCA) projections simply as a convenient visualization tool. Other than observation that in some cases the temporal/input components live in a low-dimensional subspace, none of our quantitative analysis is dependent upon the PCA projections. For all one-to-one plots, a large percentage ($> 90\%$) of the variance is explained by the first 2 or 3 PC dimensions.

[5]For $N$ possible input words, the encoder input components align with the vetrices of an $(N - 1)$-simplex, which is similar to the classification behavior observed in Aitken et al. (2020).

[6]Since in AED we pass both the decoder hidden state and the context vector to the readout, each readout vector is twice the hidden state dimension. We plot only the readout weights corresponding to the context vector, since generally those corresponding to the decoder hidden state are negligible, see SM for more details.

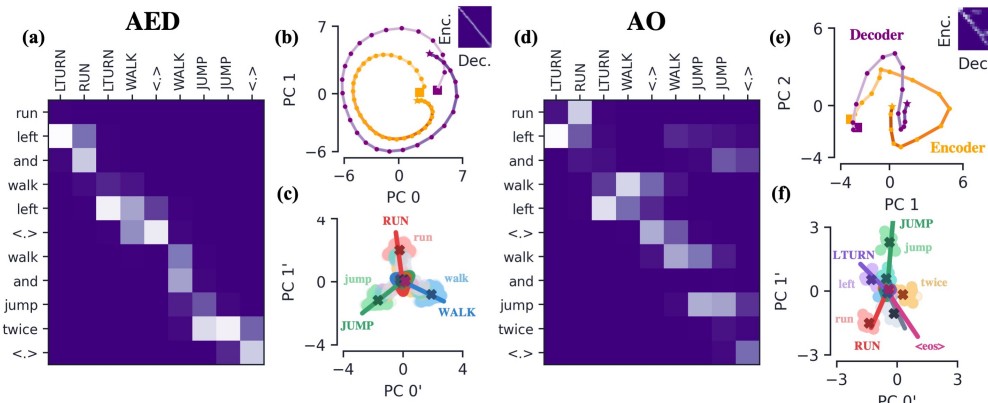

Figure 3: **Summary of dynamics for AED and AO architectures trained on eSCAN. (a)** Example attention matrix for the AED architecture. **(b)** AED network's temporal components, with the inset showing the attention matrix from said temporal components. Once again, encoder and decoder components are orange and purple, respectively and we are projecting onto the temporal component PCs. **(c)** AED network's input-delta components, input components, and readouts, all colored by their corresponding input/output words (see labels). All quantities projected onto input component PCs. **(d, e, f)** The same plots for AO.

As recurrence is the only method of transferring information across time steps, encoder and decoder hidden states must carry all relevant information from preceding steps. Together, this results in the delta components deviating significantly more from their respective input components for VED relative to AED and AO (Fig. 2f). That is, since hidden states must hold the information of inputs/outputs for *multiple* time steps, we cannot expect them to be well approximated by $\boldsymbol{\mu}_t^{\mathrm{E}} + \boldsymbol{\chi}_x^{\mathrm{E}}$ because, by definition, it is agnostic to the network's inputs at any time other than $t$ (and similarly for $\boldsymbol{\mu}_s^{\mathrm{E}} + \boldsymbol{\chi}_y^{\mathrm{E}}$). As such, the temporal and input component decomposition gains us little insight into the inner workings of the VED architecture. Additional details of the VED architecture dynamics are discussed in the SM.

**Additional Tasks.** In this section, we briefly address how two additional synthetic tasks can be understood using the temporal and input component decomposition. First, consider a task identical to the one-to-one task, with the target sequence reversed in time, e.g. $\{\mathrm{B}, \mathrm{A}, \mathrm{C}, \mathrm{A}, \mathrm{D}\} \rightarrow \{4, 1, 3, 1, 2\}$. For this task, we expect an attention matrix that is anti-diagonal (i.e. it is nonzero for $t = S + 1 - s$). For the AED and AO networks trained on this task, we find their temporal and input component behavior to be identical to the original one-to-one task with one exception: instead of the encoder and decoder temporal components following one another, we find one trajectory is flipped in such a way as to yield an anti-diagonal attention matrix (Fig. 2g). That is, the last encoder temporal component is aligned with the first decoder temporal component and vice versa.

Second, consider the task of sorting the input alphabetically, e.g. $\{\mathrm{B}, \mathrm{C}, \mathrm{A}, \mathrm{D}\} \rightarrow \{\mathrm{A}, \mathrm{B}, \mathrm{C}, \mathrm{D}\}$. For this example, we expect the network to learn an input-dependent attention matrix that correctly permutes the input sequence. Since there is no longer a correlation between input and output sequence location, the average attention matrix is very different from that of a random sample, and so we expect the temporal components to insignificantly contribute to the alignment. Indeed, we find $\boldsymbol{\mu}_s^{\mathrm{D}} \cdot \boldsymbol{\mu}_t^{\mathrm{E}}$ to be negligible, and instead $\Delta \mathbf{h}_s^{\mathrm{D}} \cdot \boldsymbol{\chi}_x^{\mathrm{E}}$ dominates the alignment values (Fig. 2h).

## 4 Beyond One-to-One Results

In this section we analyze the dynamics of two tasks that have "close-to-diagonal" attention: (1) what we refer to as the extended SCAN dataset and (2) translation between English and French phrases. Since we found temporal/input component decomposition to provide little insight into VED dynamics, our focus in this section will be on only the AED and AO architectures. For both tasks we explore below, parts of the picture we established on the one-to-one task continues to hold. However, we will see that in order to succeed at these tasks, both AO and AED must implement additional mechanisms on top of the dynamics we saw for the one-to-one task.

**Extended SCAN (eSCAN)** is a modified version of the SCAN dataset (Lake & Baroni, 2018), in which we randomly concatenate a subset of the phrases to form phrases of length 15 to 20 (see SM for

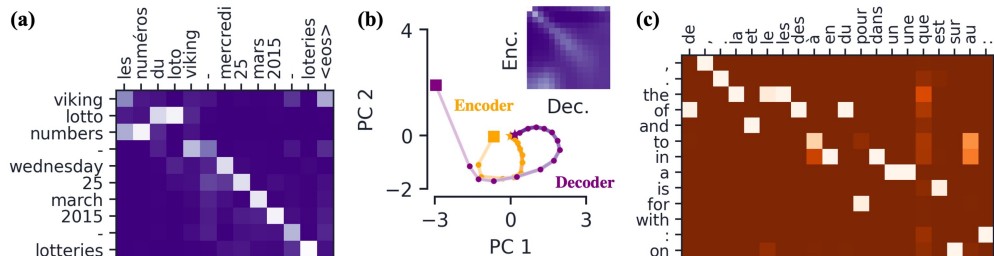

Figure 4: **Summary of features for AO trained on English to French translation. (a)** Sample attention matrix. **(b)** The encoder (orange) and decoder (purple) temporal components, with a square and star marking the first and last time step, respectively. Once again, quantities are projected onto the temporal component PCs. The inset shows the attention matrix from the temporal components, i.e. the softmax of $\boldsymbol{\mu}_s^D \cdot \boldsymbol{\mu}_t^E$. **(c)** The dot product between the most common output word readouts and the most common input word input components, $\chi_x^E$.

details). The eSCAN tasks is close to one-to-one translation, but is augmented with several additional rules that modify its structure. For example, a common sequence-to-sequence structure is that a pair of outputs can swap order relative to their corresponding inputs: the English words 'green field' translate to 'champ vert' in French (with 'field' $\leftrightarrow$ 'champ' and 'green' $\leftrightarrow$ 'vert'). This behavior is present in eSCAN: when the input word 'left' follows a verb the output command must first turn the respective direction and then perform said action (e.g. 'run left' $\rightarrow$ 'LTURN RUN').

The AED and AO models both achieve $\geq 98\%$ word accuracy on eSCAN. Looking at a sample attention matrix of AED, we see consecutive words in the output phrase tend to attend to the same encoder hidden states at the end of subphrases in the input phrase (Fig. 3a). Once again decomposing the AED network's hidden states as in (2), we find the temporal components of the encoder and decoder form curves that mirror one another, leading to an approximately diagonal attention matrix (Fig. 3b). The delta components are significantly less negligible for this task, as evidence by the fact $\chi_x^E + \Delta \mathbf{h}_t^E$ aren't nearly as clustered around their corresponding input component (Fig. 3c). As we will verify later, this is a direct result of the network's use of recurrence, since now hidden states carry information about subphrases, rather than just individual words.

Training the AO architecture on eSCAN, we also observe non-diagonal attention matrices, but in general their qualitative features differ from those of the AED architecture (Fig. 3d). Focusing on the subphrase mapping 'run twice' $\rightarrow$ 'RUN RUN', we see the network learns to attend to the word preceding 'twice', since it can no longer rely on recurrence to carry said word's identity forward. Once again, the temporal components of the encoder and decoder trace out paths that roughly follow one another (Fig. 3e). We see input-delta components cluster around their corresponding input components, indicating the delta components are small (Fig. 3f). Finally, we again see the readouts of particular outputs align well with the input components of their corresponding input word.

**English to French Translation** is another example of a nearly-diagonal task. We train the AED and AO architectures on this natural language task using a subset of the para_crawl dataset Bañón et al. (2020) consisting of over 30 million parallel sentences. To aid interpetation, we tokenize each sentence at the word level and maintain a vocabulary of 30k words in each language; we train on sentences of length up to 15 tokens.

Since English and French are syntactically similar with roughly consistent word ordering, the attention matrices are in general close to diagonal (Fig. 4a). Again, note the presence of features that require off-diagonal attention, such as the flipping of word ordering in the input/output phrases and multiple words in French mapping to a single English word. Using the decomposition of (2), the temporal components in both AED and AO continue to trace out similar curves (Fig. 4b). Notably, the alignment resulting from the temporal components is significantly less diagonal, with the diagonal behavior clearest at the beginning of the phrase. Such behavior makes sense: the presence of off-diagonal structure means, on average, translation pairs become increasingly offset the further one moves into a phrase. With offsets that increasingly vary from phrase to phrase, the network must rely less on temporal component alignments, which by definition are independent of the inputs. Finally, we see that the the dot product between the input components and the readout vectors implement the translation dictionary, just as it did for the one-to-one task (Fig. 4c, see below for additional discussion).

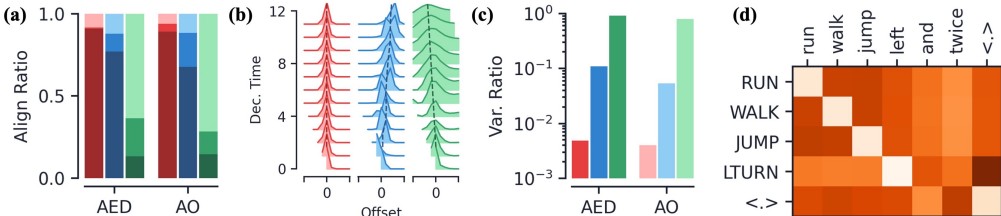

Figure 5: **Temporal and input component features.** In the first three plots, the data shown in red, blue, and green corresponds to networks trained on the one-to-one, eSCAN, and English to French translation tasks, respectively. **(a)** Breakdown of the nine terms that contribute to the largest alignment scores (see (3)) averaged across the entire decoder sequence for each task/architecture combination (see SM for details). For each bar, from top to bottom, the alignment contributions from $\boldsymbol{\mu}_s^{\mathrm{D}} \cdot \boldsymbol{\mu}_t^{\mathrm{E}}$ (dark), $\boldsymbol{\mu}_s^{\mathrm{D}} \cdot \boldsymbol{\chi}_x^{\mathrm{E}} + \boldsymbol{\mu}_s^{\mathrm{D}} \cdot \Delta\mathbf{h}_t^{\mathrm{E}}$ (medium), and the remaining six terms (light). **(b)** For the AO architecture, the dot product of the temporal components, $\boldsymbol{\mu}_s^{\mathrm{D}} \cdot \boldsymbol{\mu}_t^{\mathrm{E}}$, as a function of the offset, $t - s$, shown at different decoder times. Each offset is plotted from $[-5, 5]$ and the dotted lines show the theoretical prediction for maximum offset as a function of decoder time, $s$. Plots for the AED architecture are qualitatively similar. **(c)** For all hidden states corresponding to an input word, the ratio of variance of $\mathbf{h}_t^{\mathrm{E}} - \boldsymbol{\mu}_t^{\mathrm{E}}$ to $\mathbf{h}_t^{\mathrm{E}}$. **(d)** For AO trained on eSCAN, the dot product of input components, $\boldsymbol{\chi}_x^{\mathrm{E}}$, with each of the readouts (AED is qualitatively similar).

## 4.1 A Closer Look at Model Features

As expected, both the AED and AO architectures have more nuanced attention mechanisms when trained on eSCAN and translation. In this section, we investigate a few of their features in detail.

**Alignment Approximation.** Recall that for the one-to-one task, we found the alignment scores could be well approximated by $a_{st} \approx \boldsymbol{\mu}_s^{\mathrm{D}} \cdot \boldsymbol{\mu}_t^{\mathrm{E}}$, which was agnostic to the details of the input sequence. For eSCAN, the $\boldsymbol{\mu}_s^{\mathrm{D}} \cdot \boldsymbol{\mu}_t^{\mathrm{E}}$ term is still largely dominant, capturing $> 77\%$ of $a_{st}$ in the AED and AO networks (Fig. 5a). A better approximation for the alignment scores is $a_{st} \approx \boldsymbol{\mu}_s^{\mathrm{D}} \cdot \boldsymbol{\mu}_t^{\mathrm{E}} + \boldsymbol{\mu}_s^{\mathrm{D}} \cdot \boldsymbol{\chi}_x^{\mathrm{E}} + \boldsymbol{\mu}_s^{\mathrm{D}} \cdot \Delta\mathbf{h}_t^{\mathrm{E}}$, i.e. we include two additional terms on top of what was used for one-to-one. Since $\boldsymbol{\chi}_x^{\mathrm{E}}$ and $\Delta\mathbf{h}_t^{\mathrm{E}}$ are dependent upon the input sequence, this means the alignment has non-trivial input dependence, as we would expect. In both architectures, we find this approximation captures $> 87\%$ of the top alignment scores. For translation, we see the term $\boldsymbol{\mu}_s^{\mathrm{D}} \cdot \boldsymbol{\mu}_t^{\mathrm{E}}$ makes up a significantly smaller portion of the alignment scores, and in general we find none of the nine terms in (3) dominate above the rest (Fig. 5a). However, at early times in the AED architecture, we again see $\boldsymbol{\mu}_s^{\mathrm{D}} \cdot \boldsymbol{\mu}_t^{\mathrm{E}}$ is the largest contribution to the alignment. As mentioned above, this matches our intuition that words at the start of the encoder/decoder phrase have a smaller offset from one another than later in the phrase, so the network can rely more on temporal components to determine attention.

**Temporal Component Offset.** For the one-to-one task, the input sequence length was always equal to the output sequence length, so the temporal components were always peaked at $s = t$ (Fig. 5b). In eSCAN, the input word 'and' has no corresponding output, which has a non-trivial effect on how the network attends since its appearance means later words in input phrase are offset from their corresponding output word. This effect also compounds with multiple occurrences of 'and' in the input. The AED and AO networks learn to handle such behavior by biasing the temporal component dot product, $\boldsymbol{\mu}_s^{\mathrm{D}} \cdot \boldsymbol{\mu}_t^{\mathrm{E}}$, the dominant alignment contribution, to be larger for time steps $t$ further along in the encoder phrase, i.e. $t > s$ (Fig. 5b). It is possible to compute the average offset of input and output words in eSCAN training set, and we see the maximum of $\boldsymbol{\mu}_s^{\mathrm{D}} \cdot \boldsymbol{\mu}_t^{\mathrm{E}}$ follow this estimate quite well. Similarly, in our set of English to French translation phrases, we find the French phrases to be on average $\sim 20\%$ longer than their English counterparts. This results in the maximum of $\boldsymbol{\mu}_s^{\mathrm{D}} \cdot \boldsymbol{\mu}_t^{\mathrm{E}}$ to gradually move toward $t < s$, e.g. on average the decoder attends to earlier times in the encoder (Fig. 5b). Additionally, note the temporal dot product falls off significantly slower as a function of offset for later time steps, indicating the drop off for non-diagonal alignments is smaller and thus it is easier for the network to off-diagonally attend.

**Word Variance.** The encoder hidden states in the one-to-one task had a negligible delta component, so the hidden states could be approximated as $\mathbf{h}_t^{\mathrm{E}} \approx \boldsymbol{\mu}_t^{\mathrm{E}} + \boldsymbol{\chi}_x^{\mathrm{E}}$. By definition, $\boldsymbol{\chi}_x^{\mathrm{E}}$ is constant for a given input word, so the variance in the hidden states corresponding to a given input word is primarily contained in the temporal component (Fig. 5c). Since the temporal component is input-independent, this led to a clear understanding of how all of a network's hidden states evolve with time and input. In the AED and AO architectures trained on eSCAN, we find the variance of the input word's hidden

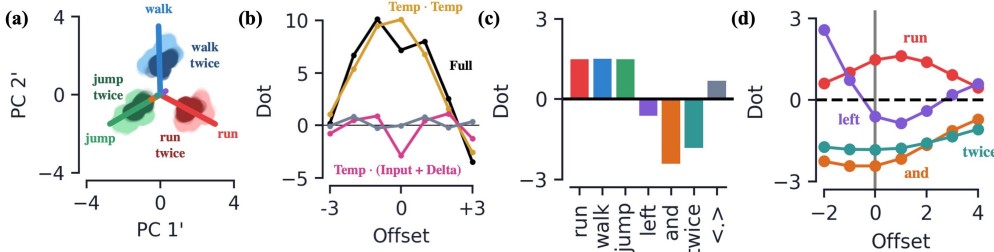

Figure 6: **How AO and AED networks implement off-diagonal attention in the eSCAN dataset. (a)** For AED, the input-delta components for various words and subphrases. **(b)** For AO, the alignment values, $a_{st}$, are shown in black when the input word 'twice' is at $t = s$. Three contributions to the alignment, $\boldsymbol{\mu}_s^D \cdot \boldsymbol{\mu}_t^E$ (gold), $\boldsymbol{\mu}_s^D \cdot \boldsymbol{\chi}_x^E + \boldsymbol{\mu}_s^D \cdot \Delta \mathbf{h}_t^E$ (pink), and $a_{st} - \boldsymbol{\mu}_s^D \cdot \mathbf{h}_t^E$ (grey) are also plotted. To keep the offset between 'twice' and the output location of the repeated word constant, this plot was generated on a subset of eSCAN with $T = S$, but we observe the same qualitative features when $T \geq S$. **(c)** The dot product between $\boldsymbol{\chi}_x^E + \Delta \mathbf{h}_t^E$ and the decoder's temporal component, $\boldsymbol{\mu}_s^D$, for $t = s$. **(d)** How the dot product of $\boldsymbol{\chi}_x^E + \Delta \mathbf{h}_t^E$ and $\boldsymbol{\mu}_s^D$ changes as a function of their offset, $t - s$, for a few select input words. The vertical gray slice represents the data in (c) and the input word colors are the same.

states drops by $90\%$ and $95\%$ when the temporal component is subtracted out, respectively (Fig. 5c). Meanwhile, in translation, we find the variance only drops by $8\%$ and $25\%$ for the AED and AO architectures, indicating there is significant variance in the hidden states beyond the average temporal evolution and thus more intricate dynamics.

**Input/Readout Alignment.** Lastly, recall we saw that in the one-to-one case the input components' alignment with readouts implemented the translation dictionary (Figs. 2b, d). For eSCAN, the dot product of a given readout is again largest with the input component of its corresponding input word, e.g. the readout corresponding to 'RUN' is maximal for the input component of 'run' (Fig. 5d). Notably, words that produce no corresponding output such as 'and' and 'twice' are not the maximal in alignment with any readout vector. Similarly, for translation, we see the French-word readouts have the largest dot product their translated English words (Fig. 4c). For example, the readouts for the words 'la', 'le', and 'les', which are the gendered French equivalents of 'the', all have maximal alignments with $\boldsymbol{\chi}_{\text{the}}^E$.

## 4.2   A Closer Look at Dynamics

In this section, we leverage the temporal and input component decomposition to take a closer look at how networks trained on the eSCAN dataset implement particular off-diagonal attentions. Many of the sequence translation structures in eSCAN are seen in realistic datasets, so we this analysis will give clues toward understanding the behavior of more complicated sequence-to-sequence tasks.

A common structure in sequence-to-sequence tasks is when an output word is modified by the words preceding it. For example, the phrases 'we run' and 'they run' translate to 'nous courrons' and 'ils courent' in French, respectively (with the second word in each the translation of 'run'). We can study this phenomenon in eSCAN since the word 'twice' tells the network to repeat the command just issued two times, e.g. 'run twice' outputs to 'RUN RUN'. Hence, the output corresponding to the input 'twice' changes based on other words in the phrase.

Since an AED network has recurrence, when it sees the word 'twice' it can know what verb preceded it. Plotting input-delta components, we see the RNN outputs 'twice' hidden states in three separate clusters separated by the preceding word (Fig. 6a). Thus for an occurrence of 'twice' at time step $t$, we have $\boldsymbol{\chi}_{\text{twice}}^E + \Delta \mathbf{h}_t^E \approx \boldsymbol{\chi}_{\text{verb}}^E + \Delta \mathbf{h}_{t-1}^E$. For example, this means the AED learns to read in 'run twice' approximately the same as 'run run'. This is an example of the network learning context.

AO has no recurrence, so it can't know which word was output before 'twice'. Hence, unlike the AED case, all occurrences of 'twice' are the same input-delta component cluster regardless of what word preceded it. Instead, it has to rely on attending to the word that modifies the output, which in this case is simply the preceding word (Fig. 3d). As mentioned in Sec. 4.1, for the eSCAN task we find the alignment to be well approximated by $a_{st} \approx \boldsymbol{\mu}_s^D \cdot \mathbf{h}_t^E$. When the word 'twice' appears in the input phrase, we find $\boldsymbol{\mu}_s^D \cdot \boldsymbol{\chi}_{\text{twice}}^E + \boldsymbol{\mu}_s^D \cdot \Delta \mathbf{h}_t^E < 0$ for $s = t$ (Fig. 6b). This decreases the value of the alignment $a_{s,s}$, and so the decoder instead attends to the time step with the second largest value

of $\boldsymbol{\mu}_s^{\mathrm{D}} \cdot \boldsymbol{\mu}_t^{\mathrm{E}}$, which the network has learned to be $t = s - 1$. Hence, $a_{s,s-1}$ is the largest alignment, corresponding to the time step before 'twice' with the verb the network needs to output again. Unlike the one-to-one case, the encoder input-delta and the decoder temporal components are no longer approximately orthogonal to one another (Fig. 6c). In the case of 'twice', $\boldsymbol{\chi}_{\mathrm{twice}}^{\mathrm{E}} + \Delta \mathbf{h}_t^{\mathrm{E}}$ is partially antialigned with the temporal component, yielding a negative dot product.

This mechanism generalizes beyond the word 'twice': in eSCAN we see input-delta components of several input words are no longer orthogonal to the decoder's temporal component (Fig. 6c). Like 'twice', the dot product of the input-delta component for a given word with its corresponding temporal component determines how much its alignment score is increased/decreased. For example, we see $\boldsymbol{\chi}_{\mathrm{and}}^{\mathrm{E}} + \Delta \mathbf{h}_t^{\mathrm{E}}$ has a negative dot product with the temporal component, meaning it leans away from its corresponding temporal component. Again, this make sense from eSCAN task: the word 'and' has no corresponding output, hence it never wants to be attended to by the decoder.

Perhaps contradictory to expectation, $\boldsymbol{\chi}_{\mathrm{left}}^{\mathrm{E}} + \Delta \mathbf{h}_t^{\mathrm{E}}$ has a negative dot product with the temporal component. However, note that the alignment of $\boldsymbol{\chi}_x^{\mathrm{E}} + \Delta \mathbf{h}_t^{\mathrm{E}}$ with the $\mathbf{h}_s^{\mathrm{D}}$ is dependent on both $t$ and $s$. We plot the dot products of $\boldsymbol{\chi}_x^{\mathrm{E}} + \Delta \mathbf{h}_t^{\mathrm{E}}$ and $\mathbf{h}_s^{\mathrm{D}}$ as a function of their offset, defined to be the $t - s$ (Fig. 6d). Notably, $\boldsymbol{\chi}_{\mathrm{left}}^{\mathrm{E}} + \Delta \mathbf{h}_t^{\mathrm{E}}$ has a larger dot product for larger offsets, meaning it increases its alignment when $t > s$. This makes sense from the point of view that the word 'left' is always further along in the input phrase than its corresponding output 'LTURN', and this offset is only compounded by the presence of the word 'and'. Thus, the word 'left' only wants to get noticed if it is ahead of the corresponding decoder time step, otherwise it hides. Additionally, the words 'and' and 'twice' have large negative dot products for all offsets, since they never want to be the subject of attention.

## 5 Discussion

In this work, we studied the hidden state dynamics of sequence-to-sequence tasks in architectures with recurrence and attention. We proposed a decomposition of the hidden states into parts that are input- and time-independent and showed when such a decomposition aids in understanding the behavior of encoder-decoder networks.

Although we have started by analyzing translation tasks, it would be interesting to understand how said decomposition works on different sequence-to-sequence tasks, such as speech-to-text. Additionally, with our focus on the simplest encoder-decoder architectures, it is important to investigate how much the observed dynamics generalize to more complicated network setups, such as networks with bidirectional RNNs or multiheaded and self-attention mechanisms. Our analysis of the attention-only architecture, which bears resemblance to the transformer architecture, suggests that a similar dynamical behavior may also hold for the Transformer, hinting at the working mechanisms behind this popular non-recurrent architecture.

## Acknowledgments and Disclosure of Funding

We thank Ankush Garg for collaboration during the early part of this work. None of the authors receive third-party funding/support during the 36 months prior to this submission or had competing interests.

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
