# A   Additional Details

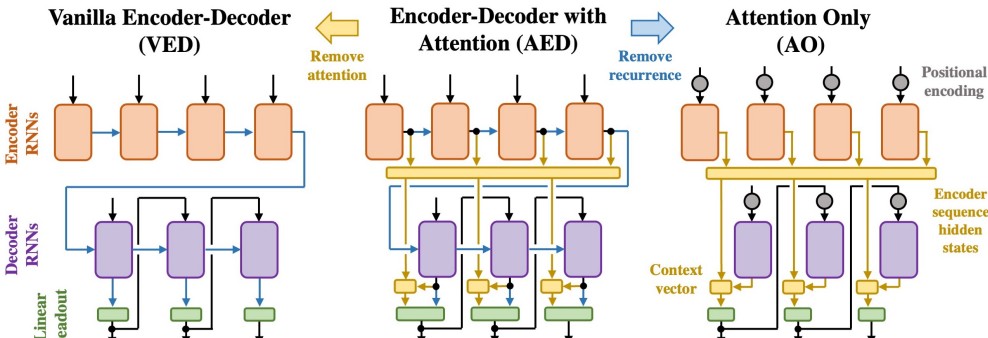

**Figure 1: Comparison of the three primary architectures used in this work and the relation between them.** The three architectures are vanilla encoder-decoder (VED), encoder-decoder with attention (AED), and attention only (AO). The encoder RNNs, decoder RNNs, and linear readout layer are showing in orange, purple, and green, respectively. Recurrent connections between RNNs are shown in blue, attention-based connections and computational blocks are shown in gold. For AO, the grey circles represent locations where positional encoding is added to the inputs. Note AED's linear readout layer takes in both the context vector from attention as well as the decoder's output.

In this section we provide additional details regarding the architectures, temporal-input component decomposition, datasets, RNNs, and training used in this work.

As a reminder, the encoder and decoder hidden states are denoted by $\mathbf{h}_t^{\mathrm{E}}$ and $\mathbf{h}_s^{\mathrm{D}}$, respectively, while inputs to the encoder and decoder are denoted by $\mathbf{x}_t^{\mathrm{E}}$ and $\mathbf{x}_s^{\mathrm{D}}$.

## A.1   Architectures

A summary of the three architectures we focus on—vanilla encoder-decoder, encoder-decoder with attention, and attention only—in this work is shown in Fig. 1. Intuitively, the architectures are related as follows: AED has attention and recurrence, VED and AO are the same as AED with the attention and recurrence removed, respectively (for AO, we also add positional encoding).

**Vanilla Encoder Decoder**

The encoder and decoder update expression are

$$\mathbf{h}_t^{\mathrm{E}} = F_{\mathrm{E}}(\mathbf{h}_{t-1}^{\mathrm{E}}, \mathbf{x}_t^{\mathrm{E}}), \qquad \mathbf{h}_s^{\mathrm{D}} = F_{\mathrm{D}}(\mathbf{h}_{s-1}^{\mathrm{D}}, \mathbf{x}_s^{\mathrm{D}}), \tag{1}$$

respectively. Here, $F_{\mathrm{D}}$ and $F_{\mathrm{E}}$ are the functions that implement the hidden state updates, which in this work are each one of three modern RNN architectures: LSTMs (Hochreiter & Schmidhuber, 1997), GRUs (Cho et al., 2014), or UGRNNs (Collins et al., 2016). The final encoder hidden state is the decoder's initial hidden state, so that $\mathbf{h}_0^{\mathrm{D}} = \mathbf{h}_T^{\mathrm{E}}$. The decoder hidden states are passed through a linear layer to get the output logits at each time step, $\mathbf{y}_s = \mathbf{W}\mathbf{h}_s^{\mathrm{D}} + \mathbf{b}$, with the following decoder input, $\mathbf{x}_{s+1}^{\mathrm{D}}$, determined by the word corresponding to the maximum output logit, $\mathrm{argmax}(\mathbf{y}_s)$.

**Encoder Decoder with Attention**

The encoder-decoder with attention architecture is identical to the VED architecture above with a simple attention mechanism added (Bahdanau et al., 2014; Luong et al., 2015). For time step $s$ of the decoder, we compute a context vector $\mathbf{c}_s$, a weighted sum of encoder hidden states,

$$\mathbf{c}_s := \sum_{t=1}^{T} \alpha_{st}\mathbf{h}_t^{\mathrm{E}}, \qquad \alpha_{st} := \frac{e^{a_{st}}}{\sum_{t'=1}^{T} e^{a_{st'}}}. \tag{2}$$

Here, $\boldsymbol{\alpha}_t := \mathrm{softmax}(a_{1t}, \ldots, a_{St})$ is the $t$-th column of the *attention matrix* and $a_{st} := \mathbf{h}_s^{\mathrm{D}} \cdot \mathbf{h}_t^{\mathrm{E}}$ the *alignment* between a given decoder and encoder hidden state. Furthermore, the outputs are now determined by passing *both* the decoder hidden state and the context vector through the linear layer, i.e. $\mathbf{y}_s = \mathbf{W}[\mathbf{h}_s^{\mathrm{D}}, \mathbf{c}_s] + \mathbf{b}$, where $[\cdot, \cdot]$ denotes concatenation Luong et al. (2015).

**Attention Only**

Attention only is identical to the AED network above, but simply eliminates the recurrent information passed from one RNN cell to the next. Since this eliminates any sense of temporal ordering in the sequences, we also add fixed positional encoding vectors (Vaswani et al., 2017), $\mathbf{p}_t^{\mathrm{E}}$ and $\mathbf{p}_s^{\mathrm{D}}$ to the encoder and decoder inputs. Together, this means the hidden state update expressions are now

$$\mathbf{h}_t^{\mathrm{E}} = F_{\mathrm{E}}(\mathbf{0}, \mathbf{x}_t^{\mathrm{E}} + \mathbf{p}_t^{\mathrm{E}}), \qquad \mathbf{h}_s^{\mathrm{D}} = F_{\mathrm{D}}(\mathbf{0}, \mathbf{x}_s^{\mathrm{D}} + \mathbf{p}_s^{\mathrm{D}}). \tag{3}$$

Note the RNN functions $F_{\mathrm{E}}$ and $F_{\mathrm{D}}$ simply act as feedforward networks in this setting. Lastly, the output logits are now determined solely from the context vector, $\mathbf{y}_s = \mathbf{W}\mathbf{c}_s + \mathbf{b}$.

Although using gated RNNs cells as feedforward networks is fairly non-standard, our primary motivation is to keep the AED and AO architectures as similar as possible in order to isolate the differences that arise from recurrence and positional encoding. Below we discuss a non-gated feedforward variant that we also briefly investigate.

Note the elimination of recurrence means the entire encoder hidden states sequence can be computed in parallel. This architecture is meant to be a simplified model of a Transformer (Vaswani et al., 2017). The hidden states output by the RNN simultaneous fill the role of the usual keys, queries, and values of a Transformer. Additionally, there is no self-attention mechanism, only a single "head" of attention, and no residual connections.

**Positional Encoding**  For a $d$-dimensional embedding dimension at time $t$, we add the vector $\mathbf{p}_t^{\mathrm{E}}$ with $i = 0, \ldots, d-1$ components:

$$p_{t,i}^{\mathrm{E}} = \begin{cases} \sin\left(\frac{t}{\tau^{i/d}}\right) & i \text{ even} \\ \cos\left(\frac{t}{\tau^{(i-1)/d}}\right) & i \text{ odd} \end{cases} \tag{4}$$

with $\tau$ some temporal scale that should be related to the phrase length. This is the same positional encoding used in Vaswani et al. (2017), and we use the same encoding for both the encoder ($\mathbf{p}_t^{\mathrm{E}}$) and decoder ($\mathbf{p}_t^{\mathrm{D}}$).

**Non-Gated Variant**  As mentioned above, we use a gated-RNN with its recurrence cut as a feedforward network. Since this is fairly non-standard, we also verify some of our results on non-gated feedforward architectures. The non-gated variant of AO uses a the following hidden-state updates

$$\mathbf{h}_t^{\mathrm{E}} = F_{\mathrm{E}}'(\mathbf{x}_t^{\mathrm{E}}) := \tanh\left(\mathbf{W}^{\mathrm{E}}\mathbf{x}_t^{\mathrm{E}} + \mathbf{b}^{\mathrm{E}}\right), \tag{5a}$$

$$\mathbf{h}_s^{\mathrm{D}} = F_{\mathrm{D}}'(\mathbf{x}_s^{\mathrm{D}}) := \tanh\left(\mathbf{W}^{\mathrm{D}}\mathbf{x}_s^{\mathrm{D}} + \mathbf{b}^{\mathrm{D}}\right), \tag{5b}$$

with tanh acting pointwise. This architecture is identical to the version of AO used above, but the hidden state updates are now

$$\mathbf{h}_t^{\mathrm{E}} = F_{\mathrm{E}}'(\mathbf{x}_t^{\mathrm{E}} + \mathbf{p}_t^{\mathrm{E}}), \qquad \mathbf{h}_s^{\mathrm{D}} = F_{\mathrm{D}}'(\mathbf{x}_s^{\mathrm{D}} + \mathbf{p}_s^{\mathrm{D}}). \tag{6}$$

Below, we show that we find the qualitative results of this network are the same as the gated version, and thus our results do not seem to be dependent upon the gating mechanisms present in the feedforward networks of the AO architecture.

## A.2   Temporal and Input Components

We define the *temporal components* to be the average hidden state at a given time step. In practice, we estimate such averages using a test set of size $M$, so that the temporal components are given by

$$\boldsymbol{\mu}_t^{\mathrm{E}} \approx \frac{\sum_{\alpha=1}^{M} \mathbf{1}_{\leq \mathrm{EoS},\alpha} \mathbf{h}_{t,\alpha}^{\mathrm{E}}}{\sum_{\beta=1}^{M} \mathbf{1}_{\leq \mathrm{EoS},\beta}}, \tag{7a}$$

$$\boldsymbol{\mu}_s^{\mathrm{D}} \approx \frac{\sum_{\alpha=1}^{M} \mathbf{1}_{\leq \mathrm{EoS},\alpha} \mathbf{h}_{s,\alpha}^{\mathrm{D}}}{\sum_{\beta=1}^{M} \mathbf{1}_{\leq \mathrm{EoS},\beta}}, \tag{7b}$$

with $\mathbf{h}_{t,\alpha}^{\mathrm{E}}$ the encoder hidden state of the $\alpha$th sample and $\mathbf{1}_{\leq \mathrm{EoS},\alpha}$ is a mask that is zero if the $\alpha$th sample is beyond the end of sentence. Next, we define the encoder *input components* to be the average

of $\mathbf{h}_t^{\mathrm{E}} - \boldsymbol{\mu}_t^{\mathrm{E}}$ for all hidden states that immediately follow a given input word (and similarly for the decoder input components). Once again, we estimate the input components using a test set of size $M$,

$$\chi^{\mathrm{E}}\left(\mathbf{x}_{t,\alpha}\right) \approx \frac{\sum_{\beta=1}^{M} \sum_{t'=1}^{T} \mathbf{1}_{\mathbf{x}_{t,\alpha},\mathbf{x}_{t',\beta}} \left(\mathbf{h}_{t',\beta}^{\mathrm{E}} - \boldsymbol{\mu}_{t'}^{\mathrm{E}}\right)}{\sum_{\gamma=1}^{M} \sum_{t''=1}^{T} \mathbf{1}_{\mathbf{x}_{t,\alpha},\mathbf{x}_{t'',\gamma}}} , \tag{8a}$$

$$\chi^{\mathrm{D}}\left(\mathbf{x}_{s,\alpha}\right) \approx \frac{\sum_{\beta=1}^{M} \sum_{s'=1}^{S} \mathbf{1}_{\mathbf{x}_{s,\alpha},\mathbf{x}_{s',\beta}} \left(\mathbf{h}_{s',\beta}^{\mathrm{D}} - \boldsymbol{\mu}_{s'}^{\mathrm{D}}\right)}{\sum_{\gamma=1}^{M} \sum_{s''=1}^{S} \mathbf{1}_{\mathbf{x}_{s,\alpha},\mathbf{x}_{s'',\gamma}}} , \tag{8b}$$

where $\mathbf{1}_{\mathbf{x}_{t,\alpha},\mathbf{x}_{t',\beta}}$ is a mask that is zero if $\mathbf{x}_{t,\alpha} \neq \mathbf{x}_{t',\beta}$ and we have temporally suppressed the superscripts on the inputs for brevity. With the above definitions, we can decompose encoder and decoder hidden states resulting from the $\alpha$th sample as

$$\mathbf{h}_{t,\alpha}^{\mathrm{E}} = \boldsymbol{\mu}_t^{\mathrm{E}} + \chi^{\mathrm{E}}\left(\mathbf{x}_{t,\alpha}^{\mathrm{E}}\right) + \Delta\mathbf{h}_{t,\alpha}^{\mathrm{E}} , \tag{9a}$$

$$\mathbf{h}_{s,\alpha}^{\mathrm{D}} = \boldsymbol{\mu}_s^{\mathrm{D}} + \chi^{\mathrm{D}}\left(\mathbf{x}_{s,\alpha}^{\mathrm{D}}\right) + \Delta\mathbf{h}_{s,\alpha}^{\mathrm{D}} , \tag{9b}$$

with the *delta components* defined to be whatever is leftover in the hidden state after subtracting out the temporal and input components,

$$\Delta\mathbf{h}_{t,\alpha}^{\mathrm{E}} := \mathbf{h}_{t,\alpha}^{\mathrm{E}} - \boldsymbol{\mu}_t^{\mathrm{E}} - \chi^{\mathrm{E}}\left(\mathbf{x}_{t,\alpha}^{\mathrm{E}}\right) , \tag{10a}$$

$$\Delta\mathbf{h}_{s,\alpha}^{\mathrm{D}} := \mathbf{h}_{s,\alpha}^{\mathrm{D}} - \boldsymbol{\mu}_s^{\mathrm{D}} - \chi^{\mathrm{D}}\left(\mathbf{x}_{s,\alpha}^{\mathrm{D}}\right) . \tag{10b}$$

In the main text we use the shorthand $\chi_x^{\mathrm{E}} = \chi^{\mathrm{E}}\left(\mathbf{x}_{t,\alpha}^{\mathrm{E}}\right)$ and $\chi_y^{\mathrm{D}} = \chi^{\mathrm{D}}\left(\mathbf{x}_{s,\alpha}^{\mathrm{D}}\right)$ (since for the decoder, the previous time step's output, $\mathbf{y}_{s-1}$ is passed as the next input). We will often suppress the batch index, so altogether the decomposition is written in the main text as

$$\mathbf{h}_t^{\mathrm{E}} = \boldsymbol{\mu}_t^{\mathrm{E}} + \chi_x^{\mathrm{E}} + \Delta\mathbf{h}_t^{\mathrm{E}} , \tag{11a}$$

$$\mathbf{h}_s^{\mathrm{D}} = \boldsymbol{\mu}_s^{\mathrm{D}} + \chi_y^{\mathrm{D}} + \Delta\mathbf{h}_s^{\mathrm{D}} . \tag{11b}$$

The intuition behind this decomposition is that it is an attempt to isolate the temporal and input behavior of the network's hidden state updates $F_{\mathrm{E}}$ and $F_{\mathrm{D}}$. This partially motivated by the fact that, *if* $F_{\mathrm{E}}$ were linear, then the encoder hidden state update for AO could be written as

$$\mathbf{h}_t^{\mathrm{E}} = F_{\mathrm{E}}(\mathbf{0}, \mathbf{x}_t^{\mathrm{E}} + \mathbf{p}_t^{\mathrm{E}}) = F_{\mathrm{E}}(\mathbf{0}, \mathbf{x}_t^{\mathrm{E}}) + F_{\mathrm{E}}(\mathbf{0}, \mathbf{p}_t^{\mathrm{E}}) . \tag{12}$$

Notably, the first term is only dependent upon the input and the second term is only dependent upon the sequence index (through the positional encoding). In this case, the temporal and input component would *exactly* capture the time and input dependence of the hidden states, respectively. Of course, $F_{\mathrm{E}}$ is not in general linear, but we still find such a decomposition useful for interpretation.

**Alignment**   Using the above decomposition, we can write the alignment as a sum of nine terms:

$$a_{st} = \left(\boldsymbol{\mu}_s^{\mathrm{D}} + \chi_y^{\mathrm{D}} + \Delta\mathbf{h}_s^{\mathrm{D}}\right) \cdot \left(\boldsymbol{\mu}_t^{\mathrm{E}} + \chi_x^{\mathrm{E}} + \Delta\mathbf{h}_t^{\mathrm{E}}\right)$$

$$= \boldsymbol{\mu}_s^{\mathrm{D}} \cdot \boldsymbol{\mu}_t^{\mathrm{E}} + \boldsymbol{\mu}_s^{\mathrm{D}} \cdot \chi_x^{\mathrm{E}} + \boldsymbol{\mu}_s^{\mathrm{D}} \cdot \Delta\mathbf{h}_t^{\mathrm{E}}$$

$$+ \chi_y^{\mathrm{D}} \cdot \boldsymbol{\mu}_t^{\mathrm{E}} + \chi_y^{\mathrm{D}} \cdot \chi_x^{\mathrm{E}} + \chi_y^{\mathrm{D}} \cdot \Delta\mathbf{h}_t^{\mathrm{E}}$$

$$+ \Delta\mathbf{h}_s^{\mathrm{D}} \cdot \boldsymbol{\mu}_t^{\mathrm{E}} + \Delta\mathbf{h}_s^{\mathrm{D}} \cdot \chi_x^{\mathrm{E}} + \Delta\mathbf{h}_s^{\mathrm{D}} \cdot \Delta\mathbf{h}_t^{\mathrm{E}} \tag{13a}$$

$$:= \sum_{I=1}^{9} a_{st}^{(I)} , \tag{13b}$$

with $a_{st}^{(I)}$ for $I = 1, \ldots, 9$ defined as the nine terms which sequentially appear after the second equality (i.e. $a_{st}^{(1)} := \boldsymbol{\mu}_s^{\mathrm{D}} \cdot \boldsymbol{\mu}_t^{\mathrm{E}}$, $a_{st}^{(2)} := \boldsymbol{\mu}_s^{\mathrm{D}} \cdot \chi_x^{\mathrm{E}}$, and so on).

In the main text, we measure the breakdown of the alignment scores from each of the nine terms. Define the contributions from one of the nine terms as

$$A_{st}^{(I)} := \frac{\left|a_{st}^{(I)}\right|}{\sum_{J=1}^{9} \left|a_{st}^{(J)}\right|} , \tag{14}$$

where the absolute values are necessary because contributions to the dot product alignment can be positive or negative.

| Input Phrase | Output Phrase |
| --- | --- |
| run $\langle.\rangle$ jump $\langle.\rangle$ walk $\langle.\rangle$ | RUN $\langle.\rangle$ JUMP $\langle.\rangle$ WALK $\langle.\rangle$ |
| run left $\langle.\rangle$ | LTURN RUN $\langle.\rangle$ |
| run twice $\langle.\rangle$ jump $\langle.\rangle$ | RUN RUN $\langle.\rangle$ JUMP $\langle.\rangle$ |
| run and jump $\langle.\rangle$ | RUN JUMP $\langle.\rangle$ |

Table 1: Extended SCAN example phrases.

### A.3 Datasets

**One-to-One Dataset**   This is a simple sequence to sequence task consisting variable length phrases with input and output words that are in one-to-one correspondence. At each time step, a word from a word bank of size $N$ is randomly chosen (uniformly), and as such there is no correlation between words at different time steps. The length of a given input phrase is predetermined and also drawn a uniform distribution. After an input phrase is generated, the corresponding output phrase is created by individually translating each word. All input words translate to a unique output word and translations are solely dependent upon the input word. An example one-to-one dataset would be converting a sequence of letters to their corresponding position in the alphabet, $\{B, A, C, A, A\} \rightarrow \{2, 1, 3, 1, 1\}$. Note the task of simply repeating the input phrase as the output is also one-to-one.

Due to the small vocabulary size, one-hot encoding is used for input phrases. Train and test sets are generated dynamically.

**Extended SCAN**   Extended SCAN (eSCAN) is a modified version of the SCAN dataset Lake & Baroni (2018). The SCAN dataset is a sequence-to-sequence task consisting of translating simple input commands into output actions. SCAN consists of roughly 20,000 phrases, with a maximum input and output phrase lengths of 9 and 49, respectively. A few relevant example phrases of eSCAN are shown in Table 1.

The eSCAN dataset modifies SCAN in two ways:

1. It takes only a subset of SCAN in which input phrases solely consist of a chosen subset of SCAN words. This allows us to isolate particular behaviors present in SCAN as well as eliminate certain word combinations that would require far-from diagonal attention mechanisms (e.g. it allows us to avoid the input subphrase 'run around thrice' that yields an output subphrase of length 24).

2. After a subset of SCAN phrases has been chosen, phrases are randomly drawn (uniformly) and concatenated together until a phrase of the desired length range is created. Individual SCAN phrases are separated by a special word token (a period). This allows us to generate phrases that are much longer than the phrases encountered in the SCAN dataset and also control the variance of phrase lengths.

The eSCAN dataset allows us to gradually increase a phrase's complexity while having control over phrase lengths. At its simplest, eSCAN can also be one-to-one if one restricts to input phrases with some combination of the words 'run', 'walk', 'jump', and 'look'.

Throughout the main text, we use the subset of SCAN that contains the words 'run', 'walk', 'jump', 'and', 'left', and 'twice' with lengths ranging from 10 to 15. Furthermore, we omit combinations that require a composition of rules, e.g. the phrase 'jump left twice' that requires the network to both understand the reversing of 'jump left' and the repetition of 'twice'. Although it would be interesting to study if and how the network learns such compositions, we leave such studies for future work. This results in 90 distinct subphrases prior to concatenation, with a maximum input and output length of 5 and 4, respectively. Post concatenation, there are over a million distinct phrases for phrases of length 10 to 15. Once again, one-hot encoding is used for the input and output phrases and train/test sets are generated dynamically.

**Translation Dataset**   Our natural-language translation dataset is the ParaCrawl Corpus, or Web Scale Parallel Corpora for European Languages (Bañón et al., 2020); we train models to translate between English and French, using the release of the dataset available in TensorFlow Datasets. This

release of ParaCrawl features 31,374,161 parallel English/French sentences; as we train our models for 30,000 steps using a batch size of 64, we do not encounter all examples during training.

To aid interpretability, we tokenize the dataset at the word level, by first converting all characters to lowercase, separating punctuation from words, and splitting on whitespace. Using 10 million randomly chosen sentences from the dataset, we build a vocabulary consisting of the 30,000 most commonly occurring words. We filter sentences which are longer than 15 tokens.

## A.4 Recurrent Neural Networks

The three types of RNNs we use in this work are specified below. $\mathbf{W}$ and $\mathbf{b}$ represent trainable weight matrices and bias parameters, respectively, and $\mathbf{h}_t$ denotes the hidden state at timestep $t$ (representing either the encoder or decoder). All other vectors ($\mathbf{c}_t, \mathbf{g}_t, \mathbf{r}_t, \mathbf{i}_t, \mathbf{f}_t$) represent intermediate quantities at time step $t$; $\sigma(\cdot)$ represents a pointwise sigmoid nonlinearity; and $f(\cdot)$ is the pointwise tanh nonlinearity.

**Gated Recurrent Unit (GRU)**   The hidden state update expression for the GRU Cho et al. (2014) is given by

$$\mathbf{h}_t = \mathbf{g}_t \cdot \mathbf{h}_{t-1} + (1 - \mathbf{g}_t) \cdot \mathbf{c}_t \,, \tag{15a}$$

with

$$\mathbf{c}_t = f\left(\mathbf{W}^{\text{ch}}(\mathbf{r} \cdot \mathbf{h}_{t-1}) + \mathbf{W}^{\text{cx}}\mathbf{x}_t + \mathbf{b}^{\text{c}}\right) \,, \tag{15b}$$

$$\mathbf{g}_t = \sigma\left(\mathbf{W}^{\text{gh}}\mathbf{h}_{t-1} + \mathbf{W}^{\text{gx}}\mathbf{x}_t + \mathbf{b}^{\text{g}}\right) \,, \tag{15c}$$

$$\mathbf{r}_t = \sigma\left(\mathbf{W}^{\text{rh}}\mathbf{h}_{t-1} + \mathbf{W}^{\text{rx}}\mathbf{x}_t + \mathbf{b}^{\text{r}}\right) \,, \tag{15d}$$

**Update-Gate RNN (UGRNN)**   The hidden state update expression for the UGRNN Collins et al. (2016) is given by

$$\mathbf{h}_t = \mathbf{g}_t \cdot \mathbf{h}_{t-1} + (1 - \mathbf{g}_t) \cdot \mathbf{c}_t \,, \tag{16a}$$

with

$$\mathbf{c}_t = f\left(\mathbf{W}^{\text{ch}}\mathbf{h}_{t-1} + \mathbf{W}^{\text{cx}}\mathbf{x}_t + \mathbf{b}^{\text{c}}\right) \,, \tag{16b}$$

$$\mathbf{g}_t = \sigma\left(\mathbf{W}^{\text{gh}}\mathbf{h}_{t-1} + \mathbf{W}^{\text{gx}}\mathbf{x}_t + \mathbf{b}^{\text{g}}\right) \,, \tag{16c}$$

**Long-Short-Term-Memory (LSTM)**   Unlike the GRU and the UGRNN, the LSTM Hochreiter & Schmidhuber (1997) transfers both a "hidden state" and a cell state from one time step to the next. In order to cast the LSTM update expressions into the same form as the GRU and UGRNN, we define its hidden state to be

$$\mathbf{h}_t = \left[\mathbf{c}_t, \tilde{\mathbf{h}}_t\right] \,, \tag{17a}$$

with the update expression given by

$$\tilde{\mathbf{h}}_t = f(\mathbf{c}_t) \cdot \sigma\left(\mathbf{W}^{\text{hh}}\mathbf{h}_{t-1} + \mathbf{W}^{\text{hx}}\mathbf{x}_t + \mathbf{b}^{\text{h}}\right) \,, \tag{17b}$$

$$\mathbf{c}_t = \mathbf{f}_t \cdot \mathbf{c}_{t-1} + \mathbf{i} \cdot \sigma\left(\mathbf{W}^{\text{ch}}\tilde{\mathbf{h}}_{t-1} + \mathbf{W}^{\text{cx}}\mathbf{x}_t + \mathbf{b}^{\text{c}}\right) \,, \tag{17c}$$

$$\mathbf{i}_t = \sigma\left(\mathbf{W}^{\text{ih}}\mathbf{h}_{t-1} + \mathbf{W}^{\text{ix}}\mathbf{x}_t + \mathbf{b}^{\text{i}}\right) \,, \tag{17d}$$

$$\mathbf{f}_t = \sigma\left(\mathbf{W}^{\text{fh}}\mathbf{h}_{t-1} + \mathbf{W}^{\text{fx}}\mathbf{x}_t + \mathbf{b}^{\text{f}}\right) \,. \tag{17e}$$

For the LSTM, we only use $\tilde{\mathbf{h}}_t^{\text{E}}$ and $\tilde{\mathbf{h}}_s^{\text{D}}$ for the determination of the context vector and as the decoder output that is passed to the readout. That is, for AED, $\mathbf{y}_s = \mathbf{W}[\tilde{\mathbf{h}}_s^{\text{D}}, \mathbf{c}_s] + \mathbf{b}$ with $\mathbf{c}_s$ the same as (2) with all $\mathbf{h}_t^{\text{E}} \to \tilde{\mathbf{h}}_t^{\text{E}}$ and the alignment $a_{st} := \tilde{\mathbf{h}}_s^{\text{D}} \cdot \tilde{\mathbf{h}}_t^{\text{E}}$.

## A.5 Training

For the one-to-one and eSCAN datasets, we train networks with the ADAM optimizer (Kingma & Ba, 2014) and an exponentially-decaying learning rate schedule with an initial learning rate of $\eta = 0.1$

and a decay rate of $0.9997$ every step (with the exception of the VED networks, in which case we used a decay rate of $0.9999$). Cross-entropy loss with $\ell_2$ regularization was used and gradients were clipped at a maximum value of 10. Both datasets used a batch size of 128 and each dynamically generated dataset was trained over two epochs. For the AED and VED architectures, a hidden dimension size of $n = 128$ was used, while for the AO we used $n = 256$ (for LSTM cells, both the hidden-state $\widetilde{\mathbf{h}}_t$ and the memory $\mathbf{c}_t$ are $n$-dimensional). For these synthetic experiments, we do not add a bias term to this linear readout layer for the purposes of simplicity and ease of interpretation. All training for these tasks was performed on GPUs and took at most 20 minutes.

As mentioned above, due to the small vocabulary size of one-to-one and eSCAN, we simply pass one-hot encoded inputs in the RNN architectures, i.e. we use no embedding layer. For the AO architecture, the input dimension was padded up 50 and 100 for the one-to-one and eSCAN tasks, respectively. Positional encoding vectors were rotated by a random orthonormal matrix so they were misaligned with the one-hot-encoded input vectors. Finally, we found performance improved when positional encoding time-scale $\tau$ was chosen to be of order the phrase length: $\tau = 50$ for the one-to-one tasks and $\tau = 100$ for the eSCAN tasks.

For the natural translation datasets, each token was mapped to an embedded vector of dimension 128 using a learned embedding layer. Both the AED and AO architectures used GRU cells, with hidden dimension size of $n = 128$. As in the synthetic datasets, we train using the ADAM optimizer for 30,000 steps using a batch size of 64; we use an exponential learning rate schedule with initial $\eta = 0.01$ and a decay rate of $0.99995$. Gradients are clipped to a maximum value of 30.

# B   Additional Results

In this section, we discuss several additional results that supplement those discussed in the main text.

## B.1   LSTM and UGRNN Cells

In the main text, all plots used a GRU RNN for the cells in the encoder and decoder. For the UGRNN and LSTM we find qualitatively similar results on the one-to-one task. Summaries of our results for the two RNN cells are shown in Fig. 2 and Fig. 3, respectively. For all types of cells, the networks achieve 100% test accuracy on the one-to-one task.

Notably, for the AED architecture, we observe that the LSTM has a slightly different attention matrix than that of the GRU and UGRNN. Namely, for the first few decoder time steps, the LSTM appears to attend to $\mathbf{h}_T^{\mathrm{E}}$ of a given phrase. This means that the network is transferring information about the first few inputs all the way to the final encoder hidden state. As such, it appears that the AED architecture with LSTM cells relies more on recurrence to solve the one-to-one task relative to its GRU and UGRNN counterparts. Similar to our findings for the VED architecture in the main text, we find the input-delta components to be significantly less clustered around their corresponding input components when this occurs (Fig. 3b). In contrast, the AO architecture with LSTM cells is qualitatively similar to that with the GRU or UGRNN cells.

We also train the AO architecture with LSTM and UGRNN cells on eSCAN. We again see qualitatively similar behavior to what we saw in the main text for AO with GRU cells (Fig. 4). For both types of cells, we see the temporal components align to form an approximately diagonal attention matrix (Fig. 4b,e). Once again, we see the input-delta components to be closely clustered around their corresponding input components, and said input components are close to their respective readouts, implementing the translation dictionary (Fig. 4c,f).

## B.2   Autonomous Dynamics and Temporal Components

When the temporal components are the leading-order behavior in alignment scores (e.g. in the one-to-one and eSCAN tasks), we find them to be well approximated the hidden states resulting from the network having zero input, i.e. $\mathbf{x}_t^{\mathrm{E}} = \mathbf{0}$ for all $t$. In the case of AED and VED, this means the encoder RNNs are driven solely by their recurrent behavior. For AO, the encoder RNNs are driven by only the positional encoding vectors. Since in all three cases this results network outputting hidden states independent of the details of the input, we call this the *autonomous* dynamics of the network.

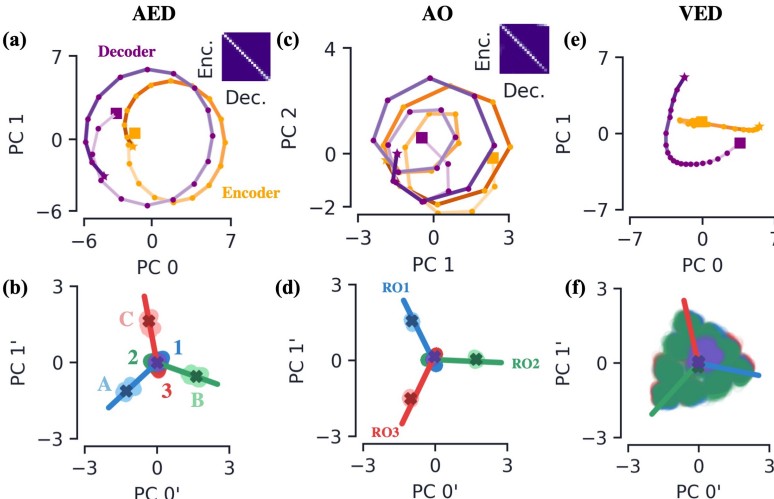

Figure 2: **Summary of dynamics for all three architectures on the one-to-one task with UGRNN cells.** All three architectures trained on an $N = 3$ one-to-one task of variable length ranging from 15 to 20. **(a)** For AED, the path formed by the temporal components of the encoder (orange) and decoder (purple), $\boldsymbol{\mu}_t^E$ and $\boldsymbol{\mu}_s^D$. We denote the first and last temporal component by a square and star, respectively, and the color of said path is lighter for earlier times. The inset shows the softmaxed alignment scores for $\boldsymbol{\mu}_s^D \cdot \boldsymbol{\mu}_t^E$, which we find to be a good approximation to the full alignment for the one-to-one task. **(b)** The input-delta components of the encoder (light) and decoder (dark) colored by word (see labels). The encoder input components, $\boldsymbol{\chi}_x^E$ are represented by a dark colored 'X'. The solid lines are the readout vectors (see labels on (d)). **(c, d)** The same plots for the AO network. **(e, f)** The same plots for the VED network (with no attention inset).

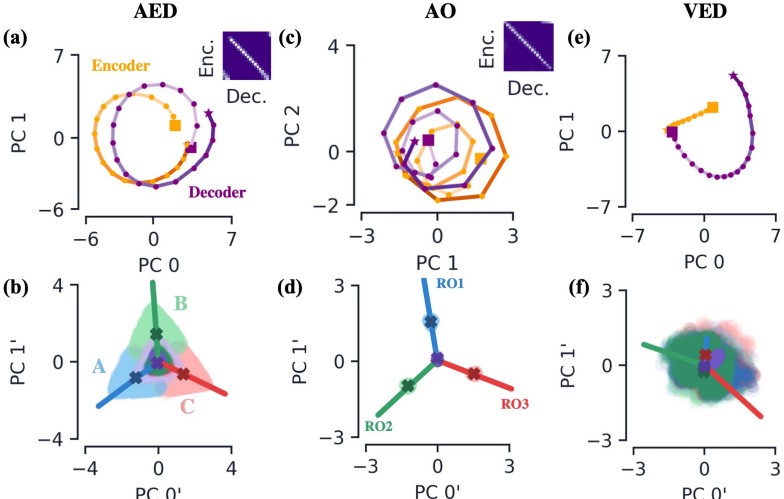

Figure 3: **Summary of dynamics for all three architectures on the one-to-one translation task with LSTM cells.** All three architectures trained on an $N = 3$ one-to-one translation task with inputs of variable length ranging from 15 to 20. **(a)** For AED, the path formed by the temporal components of the encoder (orange) and decoder (purple), $\boldsymbol{\mu}_t^E$ and $\boldsymbol{\mu}_s^D$. We denote the first and last temporal component by a square and star, respectively, and the color of said path is lighter for earlier times. The inset shows the softmaxed alignment scores for $\boldsymbol{\mu}_s^D \cdot \boldsymbol{\mu}_t^E$, which we find to be a good approximation to the full alignment for the one-to-one task. **(b)** The input-delta components of the encoder (light) and decoder (dark) colored by word (see labels). The encoder input components, $\boldsymbol{\chi}_x^E$ are represented by a dark colored 'X'. The solid lines are the readout vectors (see labels on (d)). **(c, d)** The same plots for the AO network. **(e, f)** The same plots for the VED network (with no attention inset).

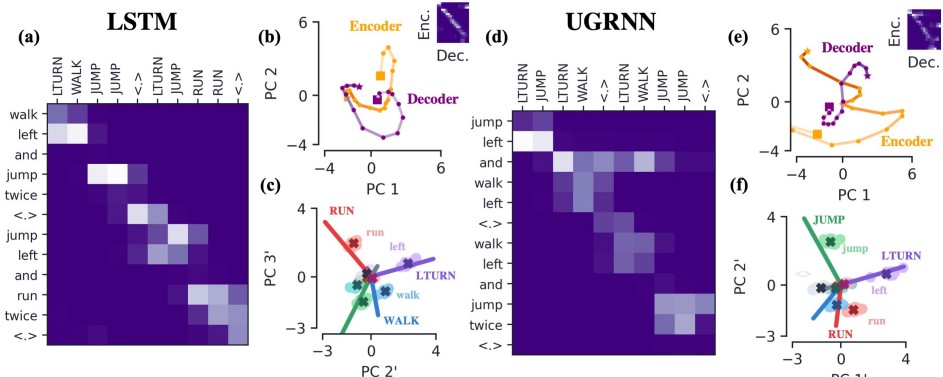

Figure 4: **Summary of dynamics for AO architectures with LSTM and UGRNN cells trained on eSCAN.**
**(a)** Example attention matrix for the AED architecture. **(b)** AED network's temporal components, with the inset showing the attention matrix from said temporal components. Once again, encoder and decoder components are orange and purple, respectively. **(c)** AED network's input-delta components, input components, and readouts, all colored by their corresponding input/output words (see labels). **(d, e, f)** The same plots for UGRNN cells.

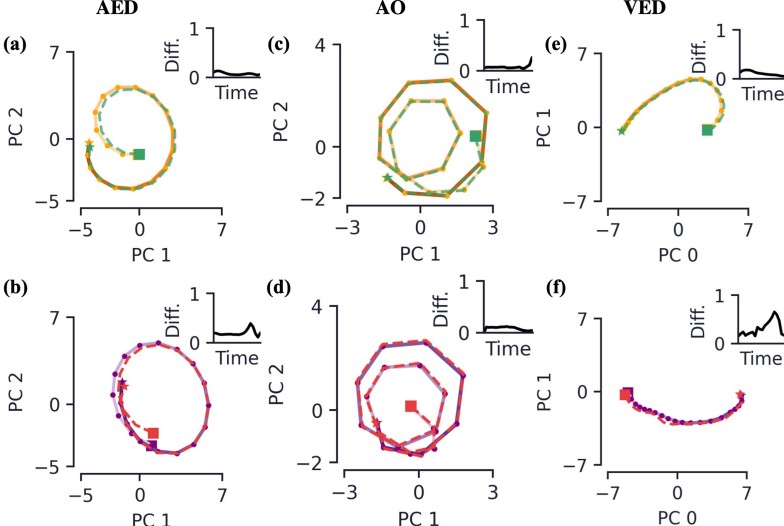

Figure 5: **Autonomous dynamics versus temporal components for architectures trained on one-to-one translation.** All three architectures are trained on an $N = 3$ one-to-one translation task with inputs of variable length ranging from 15 to 20. **(a)** For AED, the path formed by the temporal components of the encoder (orange), $\boldsymbol{\mu}_t^{\mathrm{E}}$. Also plotted in green are the null hidden states, $\mathbf{h}_t^{\mathrm{E},0}$. We denote the first and last hidden state of each of these by a square and star, respectively. The inset shows the quantitative difference between the two states, $\|\mathbf{h}_t^{\mathrm{E}} - \mathbf{h}_t^{\mathrm{E},0}\|_2 / \|\mathbf{h}_t^{\mathrm{E}}\|_2$, as a function of encoder time step, $t$. **(b)** The same plot but for the decoder temporal components (purple) and the decoder null hidden states (red). **(c, d)** The same plots for the AO network. **(e, f)** The same plots for the VED network.

Denote the encoder and decoder hidden states resulting from no input by $\mathbf{h}_t^{E,0}$ and $\mathbf{h}_s^{D,0}$, respectively. For AED and VED, they are

$$\mathbf{h}_t^{E,0} = F_E(\mathbf{h}_{t-1}^{E,0}, \mathbf{0})\,, \qquad \mathbf{h}_s^{D,0} = F_D(\mathbf{h}_{s-1}^{D,0}, \mathbf{0})\,. \tag{18}$$

For the AO network, we still add the positional encoding vectors to the inputs,

$$\mathbf{h}_t^{E,0} = F_E(\mathbf{0}, \mathbf{p}_t^E)\,, \qquad \mathbf{h}_s^{D,0} = F_D(\mathbf{0}, \mathbf{p}_s^D)\,. \tag{19}$$

Plotting the resulting hidden states along with the temporal components, we find for all three architectures the two quantities are quite close at all time steps of the encoder and decoder (Fig. 5). We quantify the degree to which the null hidden states approximate the temporal components via $\|\mathbf{h}_t^E - \mathbf{h}_t^{E,0}\|_2 / \|\mathbf{h}_t^E\|_2$ and $\|\mathbf{h}_s^D - \mathbf{h}_s^{D,0}\|_2 / \|\mathbf{h}_s^D\|_2$ where $\|\cdot\|_2$ denotes the $\ell_2$-norm. For the AO network, we find the average of this quantity across the entire encoder and decoder phrases to be about $0.07$ and $0.08$, respectively, while for AED we find it to be $0.07$ for the encoder and $0.19$ for the decoder.

We also find the null hidden states to be close to the temporal components of eSCAN (Fig. 6). Again, averaging our difference measure across the entire encoder and decoder phrases, for AO we find $0.11$ and $0.15$ and for AED $0.21$ and $0.17$, respectively.

This result gives insight into the network dynamics that drive the behavior of the temporal components. In AED and VED, each RNN cell is driven by two factors: the recurrent hidden state and the input. The AO architecture is similar, but the recurrence is replaced by temporal information through the positional encoding. The absence of input eliminates the input-driven behavior in the RNN cells. Since the network's hidden states still trace out paths very close to the temporal components, this is evidence that it is the recurrent dynamics (in the case of AED and VED) or the positional encoding vectors that drive the network's temporal component behavior.

One can use this information to postulate other behaviors in the network. For instance, given the lack of correlation between inputs in the one-to-one translation task, in the AO network one may wonder how much of the decoder's dynamics are driven by recurrent versus input behavior. We already know the decoder's primary job in this network is to align its temporal components with that of the encoder, and the results above suggest said behavior is driven primarily by the positional encoding vectors and not the inputs. To test this, we compare the accuracies of a trained network with and without inputs into the decoder RNN. We still achieve 100% word accuracy when $\mathbf{x}_s^D = \mathbf{0}$ for all decoder time steps.

### B.3  Learned Attention

In addition to the dot-product attention considered throughout the main text, we have implemented and analyzed new architectures that are identical to the AED and AO architectures that use a learned-attention mechanism. These networks use a scaled-dot product attention in the form of queries, keys, and value matrices similar to the original Transformer Vaswani et al. (2017). More specifically, the context vector $\mathbf{c}_s$ and alignment $a_{st}$ are now determined by the expressions

$$\mathbf{c}_s := \sum_{t=1}^{T} \alpha_{st} \mathbf{v}_t\,, \qquad a_{st} := \mathbf{q}_s \cdot \mathbf{v}_t\,. \tag{20}$$

In these expressions, the vectors $\mathbf{v}_t$, $\mathbf{q}_s$, and $\mathbf{k}_t$ are product of the *learned* weight matrices $\mathbf{V} \in \mathbb{R}^{n \times n}$, $\mathbf{Q} \in \mathbb{R}^{n' \times n}$, and $\mathbf{K} \in \mathbb{R}^{n' \times n}$ and the hidden states,

$$\mathbf{v}_t := \mathbf{V}\mathbf{h}_t^E\,, \qquad \mathbf{q}_s := \mathbf{Q}\mathbf{h}_s^D\,, \qquad \mathbf{k}_t := \mathbf{K}\mathbf{h}_t^E\,, \tag{21}$$

with $\mathbf{v}_t \in \mathbb{R}^n$ (i.e. the same dimension as hidden state space) and $\mathbf{q}_s, \mathbf{k}_t \in \mathbb{R}^{n'}$, with $n'$ the dimension of the query/key space.

After training these networks on our one-to-one and eSCAN tasks, we find very similar results to that of dot-product attention. In particular, temporal components of the encoder and decoder continue to align with one another after being projected through the respective key/query matrix (for AO, see Fig. 7a). Input-delta components cluster along the respective readouts, after being projected through the value matrix (Fig. 7b). Decomposing the alignment scores for these networks, we continue to find them to be dominated by the temporal component term (i.e. $\boldsymbol{\mu}_s^D \mathbf{Q}^T \mathbf{K} \boldsymbol{\mu}_t^E$).

This gives us additional confidence that the analysis techniques can be applied to modern architectures that use attention. For instance, in a multi-headed attention setting, such a decomposition could be used on each head separately to characterize the dynamics behind each of the heads.

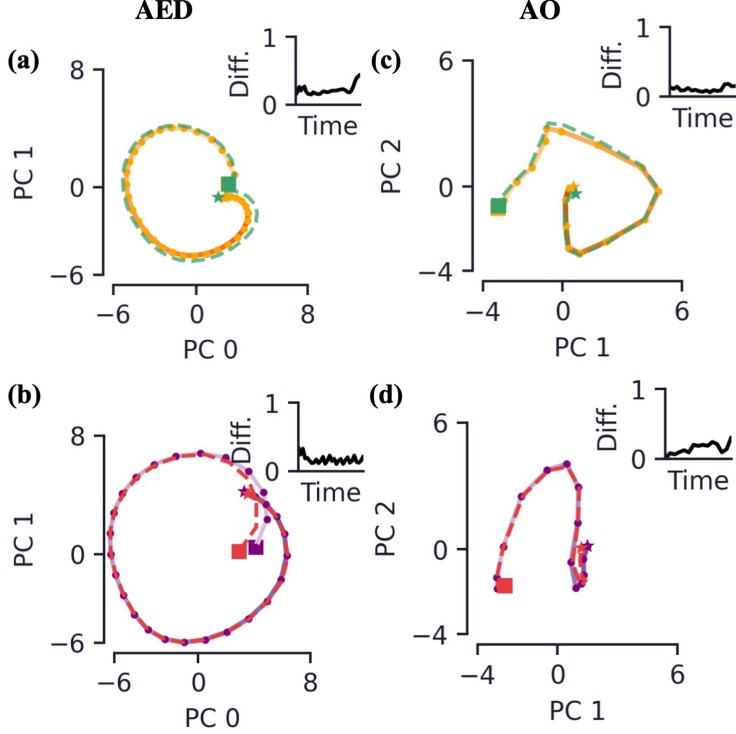

Figure 6: **Autonomous dynamics versus temporal components for architectures trained on eSCAN. (a)** For AED, the path formed by the temporal components of the encoder (orange), $\boldsymbol{\mu}_t^{\mathrm{E}}$. Also plotted in green are the null hidden states, $\mathbf{h}_t^{\mathrm{E},0}$. We denote the first and last hidden state of each of these by a square and star, respectively. The inset shows the quantitative difference between the two states, $\|\mathbf{h}_t^{\mathrm{E}} - \mathbf{h}_t^{\mathrm{E},0}\|_2 / \|\mathbf{h}_t^{\mathrm{E}}\|_2$, as a function of encoder time step, $t$. **(b)** The same plot but for the decoder temporal components (purple) and the decoder null hidden states (red). **(c, d)** The same plots for the AO network.

## B.4 Attention Only with Non-Gated Feedforward

The AO architecture's feedfoward networks are created by zeroing the recurrent part of various RNNs. Although this does result in a feedforward network, the presence of the gating mechanisms in the RNN cells we use in this work make this non-standard feedforward network. To verify our qualitative results hold beyond a gated feedfoward network, we investigated if our results differed when using a standard fully connected layer followed by a tanh readout.

We find the non-gated AO network trains well on both the one-to-one and eSCAN tasks, achieving 100% and 98.8% word accuracy, respectively. Again performing the temporal and input component decomposition on the network's hidden states, we find the qualitative dynamics of this network are the same as its RNN counterparts (Fig. 8). For example, in the one-to-one translation task we find the temporal components of the encoder and decoder again mirror one another in order to from a diagonal attention matrix (Fig. 8a). The input components of the encoder align with the readouts to implement the translation dictionary (Fig. 8b).

## B.5 Encoder-Decoder with Attention Readouts

The AED network's linear readout takes into account both the decoder hidden state output and the context vector, i.e. $\mathbf{y}_s = \mathbf{W}[\mathbf{h}_s^{\mathrm{D}}, \mathbf{c}_s]$. As such, each output's readout can be viewed as *two* vectors in hidden state space: one which acts on the context vector and the other which acts on the decoder hidden state output. Here we elaborate on our comment in the main text regarding the effects of the decoder readouts being negligible.

Recall that after training the AED network, we found the context vector readouts align with the encoder input components, yielding the translation dictionary for a given task. We find the the decoder readouts to be close to orthogonal to the context vector readouts (Fig. 9). Furthermore, we find the

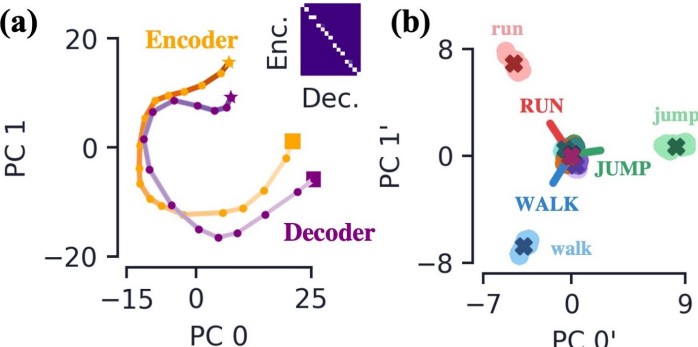

Figure 7: **Dynamics for AO with learned attention. (a)** The path formed by the temporal components of the encoder (orange) and decoder (purple) multiplied by the key and query matrices, respectively, i.e. $\mathbf{K}\boldsymbol{\mu}_t^{\mathrm{E}}$ and $\mathbf{Q}\boldsymbol{\mu}_s^{\mathrm{D}}$. We denote the first and last temporal component by a square and star, respectively, and the color of said path is lighter for earlier times. The inset shows the softmaxed alignment scores for $\mathbf{q}_s \cdot \mathbf{k}_t$, which we find to be a good approximation to the full alignment for the one-to-one translation task. **(b)** The input-delta components of the encoder (light) and decoder (dark) colored by word (see labels), after being multiplied by value matrix, $\mathbf{V}$. The encoder input components, $\mathbf{V}\boldsymbol{\chi}_x^{\mathrm{E}}$ are represented by a dark colored 'X'. The solid lines are the readout vectors.

readouts for all words other than the 'eos' character to be closely aligned. Omitting the 'eos' effect, this results in the decoder readouts contributing roughly equal values to all logit values. Since the logit values are passed through a softmax function, it is their differences that ultimately matter when it comes to classifying a given output as a word. In contrast, we found the context vector readouts to align with the vertices of an $(N-1)$-simplex and the logit value contributions to vary significantly more with the hidden state. Indeed, comparing the difference between the largest and second largest logit contribution of each set of readouts, we find the difference due to the context vector readouts to be several times larger than that of the decoder readouts. For AED trained on eSCAN, we again find the differences in the context vector logit contributions to be several times larger than that due to the decoder hidden states.

### B.6 Vanilla Encoder-Decoder Dynamics

In the main text, we briefly discussed the dynamics of the VED architecture, and here we provide some additional details. After training the VED architecture on the one-to-one translation task, we found that the encoder and decoder hidden states belonging to the same time step formed clusters, and said clusters are closest to those corresponding to adjacent time steps. Additionally, since the VED arhcitecture has no attention, the encoder and decoder hidden states have to carry all relevant information from preceding steps. To facilitate this, the dynamics of the encoder's hidden state space organizes itself into a tree structure to encode the input phrases (Fig. 10). Starting from the encoder's initial hidden state, the hidden states of a given input phrase traverse the branches of said tree as the phrase is read in, ultimately arriving at one of the tree's leaves at time $T$. The distinct leaves represent the different final encoder hidden states, $\mathbf{h}_T^{\mathrm{E}}$, that must encode the input phrase's words and their ordering.

Since the decoder receives no additional input, the encoder must place $\mathbf{h}_T^{\mathrm{E}}$ in a location of hidden state space for the decoder's dynamics to produce the entire output phrase. Although the network does indeed learn to do this, we do not observe the reversal of the tree structure learned by the encoder. That is, any two phrases that have the same output sequence for any time $s \geq s'$ could occupy the same decoder hidden states $\tilde{\mathbf{h}}_s$ for $s \geq s'$. This would result in the temporal mirror of the tree structure seen in the encoder. However, such a structure is not observed and the decoder instead seems to arrive at a solution where all output paths are kept distinct.

## References

Bahdanau, D., Cho, K., and Bengio, Y. Neural machine translation by jointly learning to align and translate. *arXiv preprint arXiv:1409.0473*, 2014.

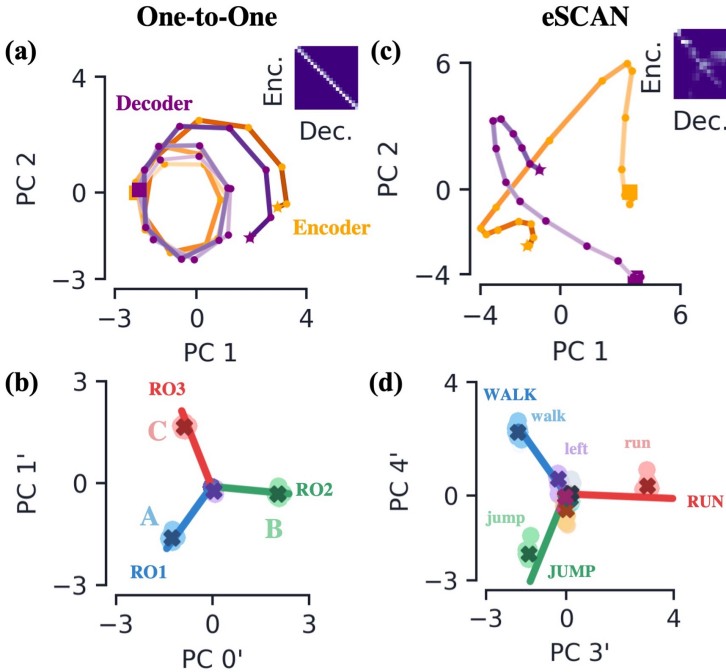

Figure 8: **AO with non-gated feed foward trained on one-to-one and eSCAN. (a)** For the AO trained on an $N = 3$ one-to-one translation task, the path formed by the temporal components of the encoder (orange) and decoder (purple), $\boldsymbol{\mu}_t^{\mathrm{E}}$ and $\boldsymbol{\mu}_s^{\mathrm{D}}$. We denote the first and last temporal component by a square and star, respectively, and the color of said path is lighter for earlier times. The inset shows the softmaxed alignment scores for $\boldsymbol{\mu}_s^{\mathrm{D}} \cdot \boldsymbol{\mu}_t^{\mathrm{E}}$, which we find to be a good approximation to the full alignment for the one-to-one translation task. **(b)** The input-delta components of the encoder (light) and decoder (dark) colored by word (see labels). The encoder input components, $\boldsymbol{\chi}_x^{\mathrm{E}}$ are represented by a dark colored 'X'. The solid lines are the readout vectors (see labels). **(c, d)** The same plots for AO trained on eSCAN.

Bañón, M., Chen, P., Haddow, B., Heafield, K., Hoang, H., Esplà-Gomis, M., Forcada, M. L., Kamran, A., Kirefu, F., Koehn, P., Ortiz Rojas, S., Pla Sempere, L., Ramírez-Sánchez, G., Sarrías, E., Strelec, M., Thompson, B., Waites, W., Wiggins, D., and Zaragoza, J. ParaCrawl: Web-scale acquisition of parallel corpora. In *Proceedings of the 58th Annual Meeting of the Association for Computational Linguistics*, pp. 4555–4567, Online, July 2020. Association for Computational Linguistics. doi: 10.18653/v1/2020.acl-main.417. URL https://www.aclweb.org/anthology/2020.acl-main.417.

Cho, K., van Merrienboer, B., Gülçehre, Ç., Bougares, F., Schwenk, H., and Bengio, Y. Learning phrase representations using RNN encoder-decoder for statistical machine translation. *CoRR*, abs/1406.1078, 2014.

Collins, J., Sohl-Dickstein, J., and Sussillo, D. Capacity and trainability in recurrent neural networks, 2016.

Hochreiter, S. and Schmidhuber, J. Long short-term memory. *Neural Computation*, 9(8):1735–1780, 1997.

Kingma, D. and Ba, J. Adam: A method for stochastic optimization. *International Conference on Learning Representations*, 12 2014.

Lake, B. and Baroni, M. Generalization without systematicity: On the compositional skills of sequence-to-sequence recurrent networks. In *International Conference on Machine Learning*, pp. 2873–2882. PMLR, 2018.

Luong, M.-T., Pham, H., and Manning, C. D. Effective approaches to attention-based neural machine translation. *arXiv preprint arXiv:1508.04025*, 2015.

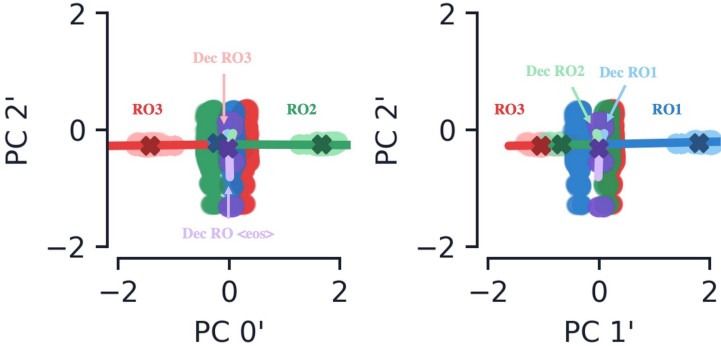

Figure 9: **Decoder hidden state behavior in AED trained on one-to-one translation.** The input-delta components of the encoder (light) and decoder (dark) colored by word. The encoder input components, $\chi_x^{\mathrm{E}}$ are represented by a dark colored 'X'. The solid lines are the readout vectors, with those corresponding to the context vector colored dark (and labeled by 'RO') and those corresponding to the decoder hidden state readout colored light (and labeled by 'Dec RO').

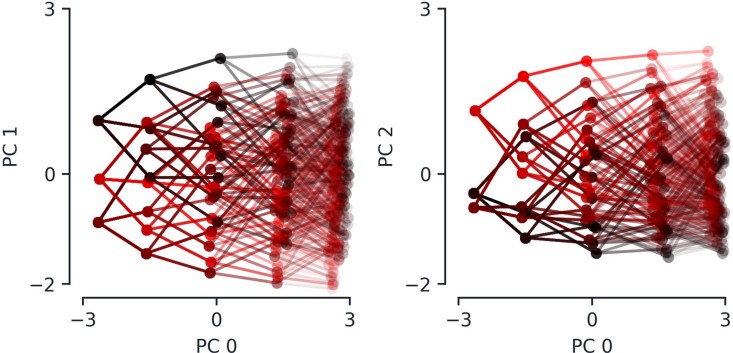

Figure 10: **Tree-like structure of encoder hidden states in VED.** The encoder hidden states of the VED architecture trained on the one-to-one translation task for the first five time steps. Note the hidden states organize themselves in five distinct clusters along PC 0. Also shown are the paths in hidden state space various input phrases take as a given input phrase is read. Said paths are colored from red to black by similarity starting from latest time, i.e. $\{B, A, C, A, C\}$ and $\{B, A, C, A, B\}$ are similarly colored but $\{C, A, C, A, C\}$ is not.

Vaswani, A., Shazeer, N., Parmar, N., Uszkoreit, J., Jones, L., Gomez, A. N., Kaiser, Ł., and Polosukhin, I. Attention is all you need. In *Advances in neural information processing systems*, pp. 5998–6008, 2017.