# OpenReview forum: "Understanding How Encoder-Decoder Architectures Attend"
_NeurIPS.cc/2021/Conference — NeurIPS 2021 Poster_

### Official Review · Reviewer_bArF · 2021-07-14

**Rating:** 6
**Confidence:** 4

**Summary:**

In this work, the authors propose a framework to analyze how different encoder-decoder architectures attend in 3 simulated tasks and 2 real-world tasks. The framework relies on isolating the  following components in the network:
1. Temporal - Average hidden state of the encoder/decoder at a specific timestep, across all the test samples.
2. Input - Average deviation of a word's hidden state from the mean state at the timepoint it occurs in, across all its occurrences in the test set.
Using these components, the paper factorizes the attention alignment between eh encoder & decoder's hidden state into 9 different products of the temporal and input states.

The usefulness of the two components in understanding attention mechanisms is demonstrated with 3 tasks:
1. Simulated tasks: Matching 3 characters to their numbered IDs | The authors find that the input and read-out states are highly aligned such that the decoder only needs to find its corresponding temporal match in the encoder. This is apparent from the results in the paper. Further, the non-attention based network doesn't demonstrate similar behavior.
2. eSCAN: Extension of the SCAN dataset formed by concatenating commands to generate longer sequences
3. English $\rightarrow$ French translation

For tasks 2 & 3, the authors find that the networks learn mappings between the input tokens and corresponding actions/French words in the readout layer. Further, the networks attend through a combination of temporal information and input components unlike task 1. Since the output in several cases doesn't depend on just 1 timepjoint in the input, these networks also exhibit 'offset' behavior wherein they attend to information at different time points depending on the input token (thus violating their orthogonality)

Overall, by separating the input and temporal components of the network state, the paper reveals if networks use only one mechanism or a combination to attend and solve a task.

**Limitations And Societal Impact:**

The authors do not discuss any limitations or societal impact of their approach.

**Main Review:**

Post discussion: I have read the author responses to both my comments and those of the other reviewers. The authors have highlighted the complexity of explaining attention mechanisms in large architectures and other real-world tasks. Overall, the proposed decomposition seems like a promising approach to study attention in encoder-decoder architectures and I would be interested in seeing how it can be used to improve model performance. I stand by my score.

###################################

Overall, I find this direction of research interesting and enjoyed reading several analyses in the paper. However, the current version of the paper lacks clarity and organization that could/should highlight main findings for each task & architecture and also highlight the usefulness of the component characterization beyond existing work (visualizing attention, attribution methods etc.). My confidence in the paper's merits would increase if the authors could point at a clear takeaway on how the networks behave. In its current form, it is hard to gauge the framework's usefulness for more complicated tasks that don't necessarily exhibit local alignment or easy 1-1 correspondence between tokens.

- While the SM contains clear definitions, lines 94-100 are confusingly worded despite being central to the paper. I’d recommend moving equations from SM to main text for readability. For instance, just from the main text it was not evident to me that $\mu_t^E$ was averaged across the test samples.
- Fig. 2:
    - There is no mention of PCA being used or why 2 PCs explain enough variance in the data that we can rely on them.
    - Why only 3 tokens?
    - What is the takeaway from the result that they traverse a circular path? It should be explicitly stated (in addition to highlighting it much later on Pg-5). I agree that they have _similar_ patterns across time, but it would be helpful to visualize in a 3D space (see Cai et al., ICLR 2021 for example) It is hard to discern if the patterns are similar because of the dimensionality or the learned structure.
    - What is the purple ‘X’ in the input component visualization?
    - There is a singular mention to ‘readout’ layers in Fig. 1 caption and no mention of it again, except directly in Fig. 2. From the footnote (and not main text) it is apparent that the readout vector is the weights of that layer on the context vector. If the readout is also a point in the space, depicting it as a line is confusing. In my opinion, it would be easier to understand the structure in the lower panels if everything was visualized as a dot/blob.
    - Line 137: It would be helpful to elaborate on why the temporal and input terms are orthogonal.
    - Overall, some of the explanation is missing/confusing, but the characterization of network behavior in Lines 140-147 is interesting!
    - Clarify the attention matrix headings for row 2 in Fig 1h.
- Fig. 3 & 4 & 5:
    - It would be helpful to demonstrate attentional differences between the two architectures in Fig. 3 by using the same input sequence.
    - Add discussion on Fig 4c to main text.
    - Overall, I think the visualizations are interesting but I’m afraid I don’t understand how they inform our understanding of the network beyond visualizing attention matrices. In other words, can the authors elaborate on what splitting the network representations into temporal and input components tells us in addition to attention matrices?
- Where are the details of this?
	> It is possible to determine the average theoretical offset of input and output words in eSCAN
- I agree with several inferences made in the real-world experiments regarding the nature of attention and the use of temporal information. But if we were looking at a task that didn’t have local alignment like in eSCAN or English$\rightarrow$French, it would be much harder to interpret under the proposed framework. As a simple example, it is not apparent to me what Fig. 4b vs. its inset indicate. One says that the temporal components are aligned in some fashion from start to end but the softmax shows that the network is relying on temporal components farther in the past as the translations get ‘offset’. What does this tell us about how the network solves the task? How is this building on our understanding beyond say, visualizing raw attention values as a heat map and averaging across samples in a test set? While these inferences fit into a clear possible explanation of the network in the simulated task, it seems a much harder ask in the English-to-French translation settings. I would appreciate if the authors can clarify the usefulness of the proposed temporal & input components framework given the intuitive results that in a non one-to-one task that relies on multi-word information like translation, the input plays a significant role and the network doesn’t rely on pure temporal alignment. Having said that, I found the characterization in Fig. 5 interesting and would be excited about future directions that can provide more insight into the model.
- I extend my comments on how interpretation may still be hard with this setup for more complex tasks. While the analysis on ‘twice’ in Section 4.2 throws light on how the input and temporal components both play a role to output required behavior (and do this differently between recurrent and position-encoding based architectures), I disagree that the analysis is equivalent to and as simple as the translation case. While _run twice_$\rightarrow$`RUN RUN` only needs to attend to the past word and copy it over, there could be examples of translation wherein there is a paraphrase/ difference in case based on meaning etc. wherein the one-to-one mapping is lost like in _we_$\rightarrow$_nous_ or _ils_$\rightarrow$_they_. To this end, what is the scope of this method?

**Time Spent Reviewing:**

5

---

> ### Author Response · Authors · 2021-08-10
> **Reply to Reviewer bArF**
>
> We thank the reviewer for the positive feedback and detailed comments on our work. Below, we respond to each point raised by the reviewer.
>
> > However, the current version of the paper lacks clarity and organization that could/should highlight main findings for each task & architecture and also highlight the usefulness of the component characterization beyond existing work (visualizing attention, attribution methods etc.). My confidence in the paper's merits would increase if the authors could point at a clear takeaway on how the networks behave.
>
> Given the increasingly widespread use of attention, we believe understanding the dynamics behind such mechanisms is a useful pursuit, even if it doesn't immediately lead to benefits that might improve network design. We believe the way attention is computed by networks is often not a well understood topic, and although we only begin to answer such questions in the simplified architectures and tasks explored in this work, we hope it lays the groundwork for future studies that might lead to more generalizable knowledge that leads to improved design and use of models that are attention-based.
>
> For example, one of the main findings of our work is that networks that have more of an average attention matrix and/or local attention are more likely to have a larger temporal component. Semantically similar languages such as English/French should use more temporal components than semantically different languages, such as English/Chinese, that may exhibit larger input/delta components.
>
> > While the SM contains clear definitions, lines 94-100 are confusingly worded despite being central to the paper. I’d recommend moving equations from SM to main text for readability.
>
> This was also pointed out by another reviewer and we have moved the formal definitions in the SM to the main text.
>
> > There is no mention of PCA being used or why 2 PCs explain enough variance in the data that we can rely on them.
>
> We have added a clarifying statement to Figures 2, 3, and 4 as well as the main text discussing the effectiveness of using two-dimensional PCA in our plots.
>
> > Why only 3 tokens?
>
> This was simply done for simplicity of visualization and explanation. Our results readily generalize beyond three tokens (see footnote 4).
>
> > What is the takeaway from the result that they traverse a circular path? It should be explicitly stated (in addition to highlighting it much later on Pg-5). I agree that they have similar patterns across time, but it would be helpful to visualize in a 3D space (see Cai et al., ICLR 2021 for example) It is hard to discern if the patterns are similar because of the dimensionality or the learned structure.
>
> The circular path of the temporal components is one particular way of keeping the temporal components of the encoder and decoder aligned. We can only hypothesize as to why the AED architectures decided to align its temporal components in such a way, but it is a particularly efficient in several ways: (1) the dynamics relating one state to the next are particularly simple, since they only require the state rotate by a constant angle, (2) the magnitude of the temporal hidden states stay relatively constant, and (3) final encoder state ends up close to the initial decoder state, requiring little change in the initial decoder dynamics. Finally, note that since the temporal components of the AO architecture are driven by the positional encoding vector, which consists of several sinusoids that oscillate as the phrase is read in, the AED is essentially learning similar (two-dimensional) sinusoidal dynamics.
>
> We tried plotting the temporal components in 3D space, and unfortunately we believe the temporal alignment is harder to visualize in these situations. Although the 2D plot doesn’t do a perfect job of representing the data, the true alignment can always be discerned from the inset of the plots.
>
> > What is the purple ‘X’ in the input component visualization?
>
> Thanks for pointing out this plotting error, that should not be there and we have updated the plots in the paper to reflect this.
>
> > There is a singular mention to ‘readout’ layers in Fig. 1 caption and no mention of it again, except directly in Fig. 2. From the footnote (and not main text) it is apparent that the readout vector is the weights of that layer on the context vector. If the readout is also a point in the space, depicting it as a line is confusing. In my opinion, it would be easier to understand the structure in the lower panels if everything was visualized as a dot/blob.
>
> We have added additional wording in the main text to remind the readers of the readouts close to the first mention of Fig. 2a. Note that for simplicity of interpretation, the linear readout layer has no bias term in it, so we felt it was useful to interpret the readout layer as the dot product of three separate vectors, hence the use of lines from the origin (to represent vectors) rather than points. Additionally, this helps the readout vectors stand out from everything else that is being plotted. We will add additional text to clarify this use and alleviate confusion.
>
> > It would be helpful to elaborate on why the temporal and input terms are orthogonal.
>
> We have added an additional sentence as to why this would be useful for the network to learn in the given task.
>
> > Clarify the attention matrix headings for row 2 in Fig 1h
>
> Clarified the caption for this subfigure.
>
> > It would be helpful to demonstrate attentional differences between the two architectures in Fig. 3 by using the same input sequence.
>
> We have changed the examples so that they are the same.
>
> > Add discussion on Fig 4c to main text.
>
> We have added a few sentences on the significance of the alignment shown in 4c.
>
> > Overall, I think the visualizations are interesting but I’m afraid I don’t understand how they inform our understanding of the network beyond visualizing attention matrices. In other words, can the authors elaborate on what splitting the network representations into temporal and input components tells us in addition to attention matrices?
>
> (Responded to this in more detail with a similar question below)
>
> > Where are the details of this?
>
> Using the word “theoretical” here was incorrect, we have corrected it to “numerical”. We have added the details of this numerical calculation to the SM. In summary, it essentially amounts to keeping track of any words that offset the positions of words in the input/output phrase (such as “left” and “and”) and numerically calculating the offset for a very large set of eSCAN phrases.
>
> > What does this tell us about how the network solves the task? How is this building on our understanding beyond say, visualizing raw attention values as a heat map and averaging across samples in a test set?
>
> We believe such a decomposition is useful toward understanding how a network is attending and, more generally, how it is operating under the hood. Additionally, it gives us insight as to how things like recurrence might change the network dynamics. The reason for spending so much time understanding the one-to-one is that it represents an extreme example where the network’s attention is only dependent upon the relative positioning of the words in the input/output phrases. In contrast, the example in Fig. 2h explores another limit, where the network’s attention is based only upon looking for a given word in the input phrase and not at all on relative positioning. In practice, the eSCAN and language translation examples lie somewhere in between these two extremes. Additionally, Section 4.1 is meant to elaborate on how one can use this analysis to dive deeper into how a network chooses to solve a task.
>
> We certainly acknowledge that there are sequence-to-sequence tasks where this decomposition may not be as useful. For instance, translation between semantically dissimilar languages such as English and Chinese may have such a dynamic attention structure where the delta components dominate, in which case further analysis is needed. Nevertheless, we believe this analysis is useful in specific domains as well as for identifying different ways a network may attend.
>
> > I disagree that the analysis is equivalent to and as simple as the translation case. While run twice → run run only needs to attend to the past word and copy it over, there could be examples of translation wherein there is a paraphrase/ difference in case based on meaning etc. wherein the one-to-one mapping is lost like in we → nous or ils → they.
>
> It is certainly true that simply copying the previous word is different from paraphrasing or having a given word be modified by preceding words. Perhaps a better analog would be along the lines of run twice → RUN SPRINT. In this case, it would seem that the network would need to partially attend to both the words “run” and “twice” rather than just the previous word. Nevertheless, we believe the attention suppressing mechanism discussed in this situation would still be useful in such cases. Perhaps, rather than the attention to “twice” being completely negated, it would be lessened such that the network similarly attends to both words. We have changed the wording around this comparison in the main text. Thank you for pointing this out.

---

> > ### Comment · Reviewer_bArF · 2021-08-24
> > **Additional questions/comments**
> >
> > Thank you for your responses!
> > 1. Re English/French vs. English/Chinese: I found this comment interesting . Perhaps adding this experiment in the future can clearly demonstrate what the different components are capturing.
> > 2. PCA: What is the amount of variance explained by the first 2 PCs and why is this an effective method?

---

> > > ### Author Response · Authors · 2021-08-30
> > > **Response to additional questions/comments**
> > >
> > > 1. Yes, we would be very interested to further analyze semantically different languages in the future and see how our methods need to be adapted to capture the more complicated translation structure that would be present in such a task. This work was primarily intended to lay the groundwork for future studies in more complicated translation structures with more sophisticated architectures.
> > >
> > > 2. Note that we use the PCA projections simply as a convenient visualization tool. That is, other than observation that in some cases the temporal/input components live in a low-dimensional space, none of our quantitative analysis is dependent upon the PCA projections. We mostly use them to qualitatively convey a general sense of the dynamics of the temporal and input components.
> > > For all the encoder-decoder temporal component plots, a large percentage (>90%)  of the variance is explained by the first 2 or 3 PC dimensions. Note that the inset of the temporal component plots show the full alignment of the temporal components, so is independent of the PCA projection.
> > > The input components of the one-to-one task (i.e. Figs. 2b, 2d, 2f) all have a >95% of variance contained in the first 2 PC dimensions. (This was the reasoning behind choosing only three words for the one-to-one plots shown in the paper. Choosing N words resulted in a roughly (N-1)-dimensional input-component space.) For the eSCAN task, the input-component space was slightly higher dimensional since there are more words in the input/output phrases. Thus the PCA plots of Figs. 3c and 3f are not meant to convey the full input-component space, just some substructure shown in said space. Plots such as Fig. 5d are meant to better convey the full alignment between input components and the respective readouts and are independent of the PCA projections. The input-component space of the English-French translation task was significantly higher dimensional, hence we did not plot its input-components.
> > > Let us know if you are interested in additional details or think the text should be modified to better convey these values.

---

> > > > ### Comment · Reviewer_bArF · 2021-09-01
> > > > **Thank you for clarifications**
> > > >
> > > > Thank you for your responses, I have no further questions!

---

### Official Review · Reviewer_GYkA · 2021-07-15

**Rating:** 5
**Confidence:** 3

**Summary:**

This paper proposes a novel method to decompose hidden representations of encoder-decoder architectures into two kinds of components: temporal and input-driven. The temporal component is the average hidden representation at a given time step thus it only varies with time, while the input-driven component is defined as the representation of a specific input word removing temporal component thus it is input-dependent but time-independent. Based on the decomposition, the authors show how encoder-decoder architectures attend, providing some insights for future research.

**Limitations And Societal Impact:**

This paper should focus more on mainstream sequence-to-sequence learning architectures and tasks.

**Main Review:**

Overall, I think the proposed decomposition method is very interesting and intuitive, which provides a new way for understanding model behaviors of (attention-based) sequence-to-sequence learning models. However, I think it still has the following weaknesses:

***1.	The studied architectures are unrepresentative.***
In this paper, the authors study the architectures of the recurrence-based encoder-decoder architecture without any attention (VED), the attention-based encoder-decoder architecture (AED), and the attention-based encoder-decoder architecture without any recurrence (AO). However, the most widely used Transformer architecture has not been studied. The authors claim that AO bears resemblance to Transformer, but I think both the analyses and conclusions will be significantly changed in Transformer due to its much stronger layer-to-layer abstraction. The analyses and conclusions on Transformer might make the proposed decomposition method more convincing and more widely used, bringing more benefits to the community. Therefore, I think that the experiments on Transformer should be indispensable. The audiences will be less interested in a paper on sequence-to-sequence learning without Transformer experiments.

***2.	The studied tasks are unrepresentative.***
This paper only studies some very simple sequence-to-sequence learning tasks, thus a question is raising that do the contributions listed in the introduction part can really inspire current research on sequence-to-sequence learning, like machine translation, question answering, and text summarization? Do the observations of the temporal and input components really exist in the more complicated task? Although the authors have conducted experiments on the English-French translation task, the uncommon settings (e.g., only training on sentences of length up to 15 tokens) make it less convincing.


**Time Spent Reviewing:**

8

---

> ### Author Response · Authors · 2021-08-10
> **Reply to Reviewer GYkA**
>
> In regards to the studied architectures and tasks being unrepresentative:
>
> We acknowledge that the focus of our work is on simplified architectures and sequence-to-sequence tasks. However, we see this as a benefit, not a drawback, of the work -- real world datasets are complicated, with multiple confounding factors influencing network dynamics. Here we have chosen tasks that elucidate particular computations and shown how networks utilize attention to realize those computations. Given the difficulty of answering the question of how a network attends, and, furthermore, how this question is more than likely domain-specific, we felt the need to approach this problem using the aforementioned two simplifications. Thus the primary goal and contributions of this paper are not meant to explain how attention works in modern architectures on a wide range of tasks but rather to propose techniques that can make progress toward such an understanding in the aforementioned class of tasks/architectures. Our hope is that this work lays a foundation for additional studies in understanding the dynamics behind attention mechanisms. It is important to start building this foundation using simple tasks and architectures where we can understand most, if not all, of the relevant network behavior.
>
> We believe understanding how the dynamics behind the Transformer attends is a very difficult task given the sheer number of separate attention layers and heads. Given the widespread use of attention-layers, better understanding its dynamics in such architectures would certainly be a significant leap forward. For this reason we have introduced the simplified AO architecture that we believe has many of the defining attributes of the Transformer, with the exception of its stacked layers (including learned attention, which is addressed in the Appendix). This makes us hopeful that our findings in the AO architecture will lend insight into these more complicated settings as well. For example, as the reviewer mentioned, the Transformer has a much stronger layer-to-layer abstraction. Perhaps our analysis would be useful in later layers where the abstractions of the earlier layers might simplify the task at hand.
>
> The reviewer does raise a very important point about whether or not the temporal and input components decomposition will generalize well to more complicated tasks, especially those further from the translation examples analysed in this work such as question answering, text summarization, or translation between semantically dissimilar languages. Based on these studies, this is an open question, which requires detailed study of the more complicated tasks.  We plan to do this in future works. However, without this study, the very decomposition would not be known, and we would not have this place to start analysis of more complicated tasks.

---

> > ### Comment · Reviewer_GYkA · 2021-08-22
> > **Regarding the English-French translation task**
> >
> > Thank you for the response. I am still curious about the uncommon setting of the English-French translation task. In your paper, the model is trained on sentences of length up to 15 tokens, but the common setting of this task is up to 80 tokens. What is the intuition behind this uncommon setting? How does the model perform with longer training instances, could you provide more details? Will the analysis and observation be changed?

---

> > > ### Author Response · Authors · 2021-08-30
> > > **Reasoning behind English-French task**
> > >
> > > The primary reasoning behind this uncommon setting is simply the fact that it was significantly easier to analyze the network’s attention on this simplified task. As is evident by the data shown in the inset of Figure 4b, the temporal component plays less and less of a part in the attention matrix as one gets further into the decoder sequence. Intuitively, we attributed this to the fact that the location of the information to translate a given word can vary significantly more if it is further along in the phrase. This effect continued when we trained our architectures on longer sequences.  Additionally, since we are using significantly simplified architectures compared to what is normally used for machine translation tasks, in order to achieve reasonable performance we felt the need to simplify the task. When we trained our architectures on the full translation task, we observed poor performance in the AO architecture. Of course, the Transformer architecture that AO is meant to mimic is able to achieve very strong performance on this task, so we attributed the poor performance to the simplicity of our architectures.
> > >
> > > We would be very interested to follow up this work by analyzing more complicated attention-based architectures on the full translation setting. As to how the analysis and observations might change in this setting, we could apply the temporal-input decomposition to each layer individually. As you mentioned in your review, the layer-to-layer abstraction plays a significant role in these more complicated architectures. Since higher layers are known to capture semantics (e.g. https://arxiv.org/abs/1905.05950), for semantically similar languages (such as English-French), we might expect more regular attention matrices thus making the temporal-input decomposition more useful at extracting an understanding of the dynamics behind attention in these later layers.
> > >
> > > However, given that our motivation in this work was to lay the groundwork for such studies in simplified architectures and tasks that are amenable to the type of analysis we performed in this work, we felt the 15 token setting was sufficient as a proof of concept (especially since this isn’t significantly off from benchmarks such as the WMT test set that usually contains phrases of 30 words on average).

---

> > > > ### Comment · Area_Chair_wtvz · 2021-08-31
> > > > **Final asessment**
> > > >
> > > > Dear Reviewer GYkA,
> > > > Please let us know if the answers provided by the authors affect your final assessment, qualitatively and quantitatively in terms of your final score.
> > > > Thanks,
> > > > The AC

---

> > > > > ### Comment · Reviewer_GYkA · 2021-08-31
> > > > > **Thank you for the response!**
> > > > >
> > > > > Dear Authors and AC,
> > > > >
> > > > > I would like to keep my score (5) unchanged. Overall, I think the proposed decomposition method is interesting but the response does not provide enough evidence for alleviating my concern about the real-world applicability. It would be much better if the authors could extent the paper with experiments on representative architectures and tasks in the future version.
> > > > >
> > > > > Best,
> > > > > Reviewer GYkA

---

### Official Review · Reviewer_pVaT · 2021-07-17

**Rating:** 6
**Confidence:** 3

**Summary:**

The paper introduces a novel method for understanding how attention works in sequence alignment problems by decomposing the hidden states into temporal and input-driven contributions.  This is applied to three different network architectures, with AO only recurrent connection, AED with both recurrent connections and attention, and AO, with only attention.  They then test on some different synthetic dataset and link the attention matrix to the hidden state decomposition and to the temporal and input component dynamics.

**Limitations And Societal Impact:**

Limitations and potential negative social impacts are addressed.

**Main Review:**

Originality: The analysis and decomposition appear to be novel, the paper is cites relevant papers and contrast their contributions to their analyses.

Impact:  Understanding how neural networks attend could have a suggest methods for improving performance and increasing interpretability of neural networks with attention.  Some of the main insights seem to be that the AED attention can be approximated by looking at the AO attention and assuming an implicit attention is present on the previous local elements in the sequence.  Another insight is that AED behaves closely to AO as opposed to behaving more like the VED, but it is not explored in more depth why this is more efficient.  The results and analysis are thorough and confirm that the networks are solving the synthetic/translation problems in the potentially expected manner given the dataset setup.  However, the insights provided by the analysis don't suggest further enhancements to how to improve attention and how networks are designed, with would increase the papers impact.

Clarity:  Paper is generally well written, but there are some minor issues with the figures.  In particular, Figures 3 and 4 would be easier to compare if the same example was used for both systems.  It needs detailed study by the reader to see the differences in attention in the systems due to needing to map the different attention system to different problems.

The presentation of the results (in particular on the synthetic sections) could end up being easier to follow if the results on the simpler architecture were introduced first, namely AO not AED, as it is a simpler system.

Line 279 is missing "with"
Line 289 in supplementary has a typo

Quality: The submission appears to be technically sound, with sufficient technical detail to replicate the experiments, and the claims are well supported.  The paper does a good job of making the different architectures comparable, and the investigation into the hidden state is well visualized in various manners in the figures.



**Time Spent Reviewing:**

3.5

---

> ### Author Response · Authors · 2021-08-10
> **Reply to Reviewer pVaT**
>
> We thank the reviewer for the thorough feedback, and very much appreciate the detailed engagement with our work. We have addressed some specific points the reviewer pointed out below.
>
> >However, the insights provided by the analysis don't suggest further enhancements to how to improve attention and how networks are designed, with would increase the papers impact.
>
> We acknowledge that we propose no improvements to existing architectures/training methods that make use of the insights gained in this work. However, given the increasingly widespread use of attention in our field, we believe furthering the understanding behind such dynamics is a useful pursuit even if it doesn't immediately lead to benefits that might improve network design. Unlike many works whose goal is to visualize and interpret the attention matrix, our motivation in this work is to understand the network dynamics behind how a network generates its attention matrix. That is, we seek to interpret how the network dynamics lead to a given attention matrix and how this might differ in recurrent and Transformer-like architectures. Although we only begin to answer such questions in the simplified architectures and tasks explored in this work, we hope it lays the groundwork for future studies that might lead to more generalizable knowledge as well as improved design and use of models that are attention-based.
>
> > Figures 3 and 4 would be easier to compare if the same example was used for both systems. It needs detailed study by the reader to see the differences in attention in the systems due to needing to map the different attention system to different problems.
>
> Would you mind clarifying what you mean by “same examples” in Figures 3 and 4? We agree that making the examples for the eSCAN dataset in Figure 3 the same for both the AO and AED architectures (i.e. Figs. 3a and 3d will be for the same example phrase) can help the reader make a more direct comparison. Figure 4 only contains data for the AO architecture trained on English to French translation.
>
> > The presentation of the results (in particular on the synthetic sections) could end up being easier to follow if the results on the simpler architecture were introduced first, namely AO not AED, as it is a simpler system.
>
> This is a very good point, especially since the AED architecture is the most complicated of architectures. Our original reasoning behind presenting AED before AO was that we felt the temporal dynamics being driven by recurrence (in the case of AED) was slightly more intuitive than being driven by positional encoding (in the case of AO). Indeed, we found it quite surprising that AO exhibited similar dynamics to AED, so we thought presenting the latter first would give the reader firmer ground to stand upon. We would be happy to change the ordering of their presentation if you think the comparatively simple architecture of AO makes understanding it first easier.
>
> We hope this sufficiently addresses the concerns about this paper from the reviewer. As these were the only concerns in an otherwise fairly positive review, we hope that you will consider increasing your score to reflect the proposed amended manuscript.

---

> > ### Comment · Reviewer_pVaT · 2021-08-25
> > **Reply to authors reply.**
> >
> > >>     Figures 3 and 4 would be easier to compare if the same example was used for both systems. It needs detailed study by the reader to see the differences in attention in the systems due to needing to map the different attention system to different problems.
> >
> > > Would you mind clarifying what you mean by “same examples” in Figures 3 and 4? We agree that making the examples for the eSCAN dataset in Figure 3 the same for both the AO and AED architectures (i.e. Figs. 3a and 3d will be for the same example phrase) can help the reader make a more direct comparison. Figure 4 only contains data for the AO architecture trained on English to French
> >
> > My mistake - this comment of course applies only to Figure 3. Figure 4c is not an attention matrix, and while the different color scheme should highlight the difference in interpretation, more exposition about the relevance of 4c would be useful (as pointed out by another reviewer).
> >
> > >>    The presentation of the results (in particular on the synthetic sections) could end up being easier to follow if the results on the simpler architecture were introduced first, namely AO not AED, as it is a simpler system.
> >
> > > This is a very good point, especially since the AED architecture is the most complicated of architectures. Our original reasoning behind presenting AED before AO was that we felt the temporal dynamics being driven by recurrence (in the case of AED) was slightly more intuitive than being driven by positional encoding (in the case of AO). Indeed, we found it quite surprising that AO exhibited similar dynamics to AED, so we thought presenting the latter first would give the reader firmer ground to stand upon. We would be happy to change the ordering of their presentation if you think the comparatively simple architecture of AO makes understanding it first easier.
> >
> > My feeling is that the other approach would be clearer for the reader, but it is acceptable in either ordering, so no need to change.
> >
> > >  We hope this sufficiently addresses the concerns about this paper from the reviewer. As these were the only concerns in an otherwise fairly positive review, we hope that you will consider increasing your score to reflect the proposed amended manuscript.
> >
> > Assuming the other reviewers comments are also addressed, I've updated my score to 6.

---

### Official Review · Reviewer_mZex · 2021-07-17

**Rating:** 6
**Confidence:** 4

**Summary:**

1. Motivation: The paper focuses on understanding the behavior of networks with attention and investigating how encoder-decoder networks solve different sequence-to-sequence tasks. To this end, the authors hypothesize that representations (i.e., hidden states) can be decomposed into "temporal" and "input" components.

2. Approach: In detail, the author presents a decomposition of the hidden states of encoder/decoder into temporal (independent of input) and input-driven (independent of sequence position) parts, where the former is used to explain the temporal behavior of the network and the latter is used to describe the input behavior.

3. Experiments: This paper conducts extensive experimental studies on several common sequence-to-sequence frameworks, including vanilla RNN, RNN with attention, and simplified Transformer networks. They study the sequence-to-sequence networks on both synthetic and real-world tasks, i.e., a synthetic task, eSCAN and English-French translation tasks, and find that the temporal components contribute more when computing the attention.

**Limitations And Societal Impact:**

Yes

**Main Review:**

Strengths:
1. The paper is well written. The studied problem is interesting and important. Considering the attention-based encoder-decoder networks have been widely used for sequence-to-sequence problems, I think this paper involves an interesting topic, aiming to understand the mechanism behind the attention-based encoder-decoder networks.

2. The paper gives a very detailed analysis of various aspects of the input and temporal components, and observes some interesting phenomena regarding how the attention patterns are formed by the hidden state dynamics under different tasks.

3. The experiments, consisting of both synthetic and real-world tasks, are extensive.


Weaknesses:
1. It's not clear what generalizable knowledge we should take away from the fact that in tasks where attention *can* be computed without looking at input content, attention *is* computed without looking at input content.

2. The three tasks used for experiments and analyses are relatively simple. Most real-world tasks are not like this.
- First, among the three tasks, the English-French phrase translation is the most complex, but most experimental analyses are concentrated on the eSCAN task. It's interesting to see the analysis of hidden state dynamics can discover that the attention for the one-to-one task is only driven by temporal components, and for eSCAN, some input components such as "twice" also contribute to the formation of the attention. But it is not surprising given the simplistic nature of these two tasks, and it is not clear what indication it might have over more realistic tasks. It would be better if more analysis is done on the phrase translation tasks (especially the sequence-to-sequence networks are applied to machine translation tasks mostly), and some discussion on how the analysis could generalize to other tasks would be nice. Therefore, I recommend the authors concentrate more on the analyses of more MT tasks in addition to the English-French phrase translation.
- Second, for the English-French phrase translation task, the delta component of the hidden states becomes relatively large. I am concerned that the validity and usefulness of the "input"/"temporal" analysis if the delta dominates the hidden states when applied to more realistic tasks. Besides, I suggest the authors report BLEU scores for the translation task so that readers can check and understand the performance of the trained translation models.
- More analyses on the complicated tasks would significantly add support to your claims.

3. Although extensive experimental results are provided, it is hard for me to conclude the main contribution of this paper. The authors do have summed up some conclusions, such as the temporal component contributes more to the attention, which, however, does not lead to some deeper insights. For example, I would expect that the findings of this paper could be utilized to improve the sequence-to-sequence networks and promote their performance. The paper only provides some intuitive conclusions to the sequence-to-sequence networks, without taking a step forward and proposing some novel improvements to them. Can we get any benefits from the observation, e.g., guiding us to design better models?

4. Some parts of the paper can be made clearer for easier reading.
- The definition of temporal and input components is essential in understanding the paper, and it should be in the main paper, not in the supplementary material.
- The axis in Figure 2 should be explained in the paper, even if the reader might be able to guess its meaning.
- It would be better if there is an outline paraphrase at the end of the first section.
- The authors should report BLEU scores for the translation task so that readers can check and understand the performance of the trained translation models.

Overall:
I think this is a promising start, but the paper would benefit from both a clearer statement of the generalizable machine learning knowledge, fewer monotonic tasks, and a more detailed discussion of how to interpret findings with large \Delta. Considering the new insights and observations, I would like to give a positive suggestion for this paper.

**Time Spent Reviewing:**

6 hours

---

> ### Author Response · Authors · 2021-08-10
> **Reply to Reviewer mZex**
>
> We thank the reviewer both for the positive assessment of our work and the critical feedback on the writing. Below, we respond to each weakness point raised by the reviewer.
>
> > It's not clear what generalizable knowledge we should take away from the fact that in tasks where attention can be computed without looking at input content, attention is computed without looking at input content.
>
> We respond to this question below in our response to point 3.
>
> > The three tasks used for experiments and analyses are relatively simple. Most real-world tasks are not like this…
>
> We agree that it would be very interesting to more thoroughly investigate the dynamics behind attention for networks trained on more complicated tasks such as English-French translation. Our aim was to provide a foundation for these studies using tasks in which we could easily imagine a mechanism that the learned network could settle upon to solve the task. By looking for dynamical mechanisms that exist in these tasks that we understand well, we can then reveal the mechanisms networks are likely to learn in order to solve them. This gives us a foothold when we later investigate whether these mechanisms are present in more complicated networks, and, if not, what other mechanisms can be found. As such, we hope this work lays the groundwork for future research that wants to investigate the dynamics behind attention in both more complicated architectures as well as more complicated tasks. Like the reviewer mentions, this type of analysis may indeed have limited scope in tasks that increasingly become dominated by the delta component, but we believe better understanding attention in the subset of tasks where this is not the case is nonetheless a useful step forward.
>
> > Although extensive experimental results are provided, it is hard for me to conclude the main contribution of this paper…
>
> In response to both points 1. and 3. above: We completely acknowledge that in this paper, we propose no improvements to existing architectures/training methods that make use of the insights gained in this work. However, given the increasingly widespread use of attention layers, we believe furthering the understanding of the dynamics behind such mechanisms is a useful pursuit, even if it doesn't immediately lead to improved network design. The way attention is used by networks to solve tasks is often not a well understood topic, and although we only begin to answer such questions in the simplified architectures and tasks explored in this work, we hope it lays the groundwork for future studies that might lead to more generalizable knowledge as well as improved design and use of models that are attention-based.
>
> > Some parts of the paper can be made clearer for easier reading.
>
> Thanks for pointing these out, we added several changes in hope of making the paper clearer, including the following. We agree that moving the definition of temporal and input components to the main text makes more sense given their importance to the paper and have done so in the latest draft. We have added additional text to both the main text and figure captions explaining the axes in Fig. 2 (as well as in the captions of Figs. 3 and 4). We have added a short outline of the paper at the end of the introduction.

---

### Decision · Program_Chairs · 2021-09-27

**Decision:**

Accept (Poster)

**Comment:**

Three of four reviewers rated this paper as a 6, one reviewer still assesses this paper as being below threshold.
The concerns that remain relate to the general applicability of the method, but I think the work is in good enough shape for a poster.
The AC recommends accept as poster.